# Open LLMs are Necessary for Current Private Adaptations and Outperform their Closed Alternatives

**Vincent Hanke, Tom Blanchard, Franziska Boenisch,**
**Iyiola E. Olatunji, Michael Backes, Adam Dziedzic**[*]
CISPA Helmholtz Center for Information Security

## Abstract

While open Large Language Models (LLMs) have made significant progress, they still fall short of matching the performance of their closed, proprietary counterparts, making the latter attractive even for the use on highly *private* data. Recently, various new methods have been proposed to adapt closed LLMs to private data without leaking private information to third parties and/or the LLM provider. In this work, we analyze the privacy protection and performance of the four most recent methods for private adaptation of closed LLMs. By examining their threat models and thoroughly comparing their performance under different privacy levels according to differential privacy (DP), various LLM architectures, and multiple datasets for classification and generation tasks, we find that: (1) all the methods leak query data, i.e., the (potentially sensitive) user data that is queried at inference time, to the LLM provider, (2) three out of four methods also leak large fractions of private training data to the LLM provider while the method that protects private data requires a local open LLM, (3) all the methods exhibit lower performance compared to three private gradient-based adaptation methods for *local open LLMs*, and (4) the private adaptation methods for closed LLMs incur higher monetary training and query costs than running the alternative methods on local open LLMs. This yields the conclusion that, to achieve truly *privacy-preserving LLM adaptations* that yield high performance and more privacy at lower costs, taking into account current methods and models, one should use open LLMs.

## 1 Introduction

Recently, there has been the trend of releasing open Large Language Models (LLMs), such as LLama [20, 57], Vicuna [10], or Mistral [27] as an alternative to their proprietary closed counterparts, such as GPT from OpenAI [2], Claude from Anthropic [4], or Gemini from Google [54]. Despite the significant progress in improving open LLMs, they are still outperformed in multiple tasks by closed LLMs [11], making the latter attractive even for learning tasks from highly *private* data.

Since it was shown that private data can leak from the adaptations of LLMs [15, 16], in the last few months alone, an array of new methods for privacy-preserving adaptation of closed LLMs has been proposed by the machine learning community at multiple conferences (NeurIPS'23 [15] and ICLR'24 [25, 53, 60]). Given the lack of access to the closed LLMs parameters—which renders parameter-tuning based adaptations infeasible—they all rely on the generation of privacy-preserving discrete prompts. We detail their operational setup in Figure 1 (left).

In this work, we ask the simple yet impactful question of whether these efforts actually lead into the right direction towards the goal of achieving *truly privacy-preserving LLM adaptations*. Therefore, we thoroughly analyze the proposed methods both conceptually and empirically and compare them to

---

[*]Correspondence to adam.dziedzic@cispa.de

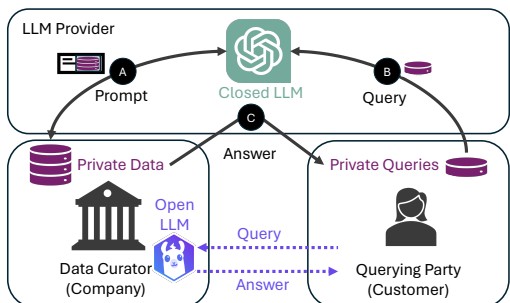

| Method | Ⓐ | Ⓑ | Ⓒ | Open LLM |
|---|---|---|---|---|
| DP-ICL [60] | ✗ | ✗ | ✓ | Not Needed |
| PromptPATE [15] | ✗ | ✗ | ✓ | Not Needed |
| DP-FewShotGen(1) [53] | ✗ | ✗ | ✓ | Not Needed |
| DP-FewShotGen(2) [53] | ✓ | ✗ | ✓ | Needed |
| DP-OPT [25] | (✓) | ✗ | ✓ | Needed |
| Private Local Adaptation | ✓ | ✓ | ✓ | Needed |

Figure 1: **Setup for Privacy Protection with Open vs Closed LLMs**. The three parties involved are (1) an LLM provider who hosts the proprietary LLM, (2) a data curator, such as a company that curated private data, for example, of their customers' previous transactions, and (3) a querying party, *i.e.,* a customer of the company who wants to perform a new private transaction. There are three steps where privacy leaks: Ⓐ During the creation of the discrete prompt, the data curator's private data leaks to the LLM provider. Ⓑ The private query of the querying party leaks to the LLM provider. Ⓒ Private information from the data curator leaks to the querying party through the returned answers of the prompted LLM [16]. Prior methods for closed LLMs [15, 53, 60] only provide protection against Ⓒ . None of them protects against Ⓑ . To prevent leakage through Ⓐ , they require access to a (powerful) local open LLM. As an alternative (dashed purple lines), the data curator could privately adapt the open LLM locally and let the querying party interact with this LLM, protecting against Ⓐ , Ⓑ , Ⓒ .

alternatives that rely on privately adapting *open local LLMs*. In particular, we study each approach's threat space, assumptions, and methodological limitations and perform extensive experiments using ten state-of-the-art open and closed LLMs of various sizes, including Vicuna, Llama 3, Open LLaMa, BERT, RoBERTa, the Pythia suite of models, Claude, two versions of GPT3 (Babbage and Davinci), and GPT4 Turbo —applied to multiple datasets both for classification and generation tasks. Our analyses cover the axes of privacy protection, performance in terms of privacy-utility trade-offs, and monetary costs for training and queries.

Our results provide the following insights: (1) All current methods for adapting closed LLMs leak private query data (intended for the data owner) at inference time to the LLM provider. (2) Three out of the four methods studied also leak large fractions of the private training data to the LLM provider. The approaches that do not, require an additional locally deployed open LLM for prompt engineering. (3) All methods for closed LLMs yield lower final downstream performance than privacy-preserving local adaptations on open LLMs—even when the local methods rely on significantly smaller LLMs than their closed counterparts. (4) The training and query costs of the private adaptations of closed LLMs (API access costs imposed by the LLM provider) are significantly higher than the costs for private open LLM adaptations (estimated as the costs of training and querying on cloud-based hardware). We provide a condensed summary of our results in Figure 1 (right Table above), and Table 1.

Overall, our results indicate that, from the perspective of effective privacy-preservation, current adaptations of open LLMs are strictly preferable over their closed LLM alternatives, since they are more private, more performant, and less expensive. Going beyond the concrete existing methods studied [15, 25, 53, 60], we then analyze the reasons behind the underwhelming results of privacy-preserving closed LLM adaptations and discuss potential directions for improvements.

On the way, to further strengthen private adaptations for open LLMs, we demonstrate how to locally apply privacy-preserving prompt-based methods to train generation tasks with high-performance— claimed impossible by prior work [35]. In particular, we show for the first time that private prompt tuning for text generation tasks PromptDPSGDGen can achieve comparable performance to private (full) fine-tuning and private low-rank adaptations (LoRA). Additionally, we demonstrate that ensemble-based few-shot prompts PromptPATEGen can privately generate high-quality text at a low privacy cost.

In summary, we make the following contributions:

Table 1: **Comparison of privacy protection, performance, and cost between private adaptations for closed vs open LLMs.** We select the top-performing adaptations. For closed LLMs, we use DP-ICL [60] and leverage PrivateLoRA [61] on open LLMs for both tasks. We consider sentiment classification on SST2 and the dialog summarization on SAMSum. The training data is denoted by $\mathcal{D}_T$ and the test queries by $Q$. *Reveals* represents which data are exposed to the LLM provider. The methods were trained with DP guarantees: $\varepsilon = 8$ and $\delta = 1/N$, where $N$ is the number of examples in $\mathcal{D}_T$. We report the *Performance* (higher is better) on test data (where *Acc* denotes the classification accuracy). The cost (in $) is computed separately for training (Train) and for answering 10k test queries (Query). Note, the (estimated) number of parameters for closed LLMs is 1.76T for GPT4 Turbo and 175B for GPT3 Davinci, while Llama3 has only 8B and BART-Large is significantly smaller with 355M parameters. The adaptation of the open LLMs is more expensive on SST2 than on SAMSum due to the larger training data size for SST2 and a larger model. DP-ICL's query cost is high due to the usage of an ensemble of 100 prompts to answer each query. *In summary, open local LLM adaptations are more private, more performant, and less expensive.*

| Adaptation | LLM Type | Model | Task | Reveals | Performance↑ | Train($) | Query($) |
|---|---|---|---|---|---|---|---|
| DP-ICL [60] | Closed | GPT4 Turbo | SST2 | $\mathcal{D}_T+Q$ | Acc=$95.9_{\pm 0.1}$% | 0 | 138.00 |
| PrivateLoRA [61] | Open | Llama3-8B(instruct) | SST2 | *None* | Acc=$96.0_{\pm 0.1}$% | 27.60 | 0.78 |
| DP-ICL [60] | Closed | GPT3 Davinci | SAMSum | $\mathcal{D}_T+Q$ | RougeL=$31.8_{\pm 0.3}$ | 0 | 665.91 |
| PrivateLoRA [61] | Open | BART-Large | SAMSum | *None* | RougeL=$39.1_{\pm 0.2}$ | 3.63 | 0.80 |

1. We perform a thorough conceptual and experimental study on existing privacy-preserving closed and open LLM adaptations, analyzing their threat space, assumptions, and achieved results.

2. Our extensive experiments on various open and closed LLMs and on multiple classification and generation tasks show that the local (gradient-based) adaptations outperform their current closed (discrete prompt-based) counterparts in terms of privacy, performance, and cost efficiency.

3. We propose differentially private prompts for text generation tasks that, for the first time, reach performance comparable to private LoRA or private fine-tuning.

## 2   Background and Related Work

**Differential Privacy.** Differential Privacy (DP) [18] is a mathematical framework that provides privacy guarantees by implementing the intuition that an algorithm $\mathcal{A} : I \rightarrow R$, executed on two neighboring datasets $D$, $D'$ that differ in only one data point (we adopted the definition of *neighboring* based on addition/removal. [35, 44]), will yield approximately the same output, *i.e.,* $\Pr[\mathcal{A}(D) \in R] \leq e^\epsilon \cdot \Pr[\mathcal{A}(D') \in R] + \delta$. While $\varepsilon$ specifies by how much the output can differ, $\delta$ specifies the probability of failure. There are two prevalent DP algorithms for training machine learning models. The first one is the differential private stochastic gradient descent algorithm (**DPSGD**) [3] where the impact of each private training data point is limited during training through gradient clipping, and privacy guarantees are integrated through the addition of calibrated amounts of stochastic noise. The second algorithm is the private aggregation of teacher ensembles (**PATE**) [48] where first, an ensemble of teacher models is trained on disjoint subsets of the private data, and then a noisy knowledge distillation is performed to a student model using public data. Another general mechanism for implementing DP is the exponential mechanism (**EM**) [43]. The EM selects an output $r$ from a set of possible outputs based on a scoring function $q(D, r)$ that measures the quality of $r$ for dataset $D$. Let $\Delta q$ be the sensitivity of the scoring function. The EM chooses $r$ with probability proportional to $\exp\left(\frac{\epsilon q(D,r)}{2\Delta q}\right)$.

**LLM Adaptations.** LLMs are pre-trained on large amounts of public data and then adapted to downstream tasks using private data [24]. We divide existing methods for private LLM adaptations into *private tuning* methods that rely on access to the LLM gradients, and *private in-context learning* (ICL) which requires only API (black-box) access to the LLM. While private tuning is only applicable to open LLMs, private ICL can, in principle, be applied to both open and closed LLMs. We note that all private LLM adaptations rely in their core on the three DP algorithms introduced above and summarize existing methods, their setup, and their assumptions in Table 2.

**Private Tuning for Open LLMs.** There exist three main ways for private tuning. **1) Prompt-based adaptations** adds a small number of parameters (usually <1% of the total number of parameters)

Table 2: **Comparison of properties between private LLM adaptations.** The in-context learning (ICL) optimizes instructions and shots (demonstrations). Many privacy techniques include the ones designed for multi-label PATE (denoted as MLPATE) [64], exponential mechanism (EM) [43], joint exponential mechanism (JEM) [21], Gaussian Mechanism (GM), Report-Noisy-Max Mechanism (RNM), Propose-Test-Release (PTR) [19], sample-and-aggregate (SAA) [46], Limited Domain Algorithm (LDA) [17].

| Property / Adaptation | Privacy Algorithms | Optimization Strategy | Privatize | Inference Type | Require |
|---|---|---|---|---|---|
| PromptDPSGD [15] | DPSGD | Gradient-based | Soft Prompt/Prefix | Multi-task | Open LLM |
| PrivateLoRA [61] | DPSGD | Gradient-based | Added parameters | Single-task | Open LLM |
| DP-FineTune [35] | DPSGD | Gradient-based | all LLM parameters | Single-task | Open LLM |
| DP-ICL [60] | RNM,GM,JEM,PTR,MLPATE | ICL | Answers | Limited Queries | None |
| PromptPATE [15] | PATE | ICL | Shots | Multi-task | Public Data |
| DP-FewShotGen [53] | GM,RNM,EM | ICL | Shots | Multi-task | Public Labels,Open LLM |
| DP-OPT [25] | SAA,LDA | ICL | Instructions+Shots | Multi-task | Validation Data,Open LLM |

only in the model input space, either on the level of token embeddings (soft prompts [39, 40]), or also to every LLM layer (prefix-tuning [30, 33]). Duan et al. [15] presented **PromptDPSGD**, which adapts the DPSGD algorithm to soft prompts. The main advantage of prompt-based adaptations is that they enable multi-task batch processing, *i.e.,* many soft prompts for different users and tasks can be processed in the same mini-batch during LLM training or inference. **2) Parameter efficient fine-tuning-based adaptations** such as LoRA [26] add a relatively small number of parameters (<10% of total number of parameters) within the model, usually in each block of a transformer architecture [58]. These added parameters are then tuned while the pre-trained original parameters remain frozen. **PrivateLoRA** [61] extends LoRA with DP guarantees by building on the DPSGD algorithm. **3) Full fine-tuning-based adaptations** either fine-tune the whole model or only a few last layers. The **DP-FineTune** [35], again based on the DPSGD algorithm, shows that full fine-tuning with DP optimization can provide strong privacy guarantees and good performance. The general trend, when choosing an adequate method, suggests that the more difficult the task, the higher the number of adaptation parameters required [15]. Thus, for simple downstream tasks, PromptDPSGD [15] is sufficient, while DP-LoRA [61] is recommended for medium-difficulty tasks, and the full fine-tuning [35] for complex tasks.

**Private ICL for Closed LLMs.** Recently, many new methods were proposed for private in-context learning with closed LLMs. All of them leverage discrete (hard) prompts and rely on a voting mechanism for privacy protection, similar to PATE [47, 48] and CaPC [12]. We divide the existing methods into the following four categories: **(1) Private Question Answering:** The work on **DP-ICL** [60] proposed to answer queries based on the private dataset. Following the PATE setup, the private data is divided into non-overlapping partitions and then each partition is prepended with an instruction to form a private teacher prompt. The prompts form an ensemble of private teachers (prompted LLMs). Since DP-ICL does not implement the idea of a student model from PATE, all the teachers (usually 100) are required to answer each query, rendering the method expensive when executed on a closed LLM. Moreover, each query incurs additional privacy cost, such that the method can answer only a limited number of queries for a given privacy budget. **(2) Private Student Prompt:** **PromptPATE** [15] tackles the problem of the high costs and the limited number of answered queries in DP-ICL by creating a student prompt. PromptPATE uses an ensemble of teacher prompts (usually around 200) to label public data. Then it selects the most performant shots for the student prompt from these newly labeled examples. **(3) Private Prompt Generation: DP-FewShotGen** [53] is similar to PromptPATE but eschews the assumption about the public data for labeling and, in turn, starting from a public label, generates each output token privately to obtain a private shot. **(4) Private Prompt Engineering:** Finally, **DP-OPT** [25] privatizes prompt engineering based on the Deep Language Network (DLN) method [51]. While DP-ICL, PromptPATE, and DP-FewShotGen assume a generic instruction and emphasize the protection of the direct leakage from the shots only, DP-OPT [25] proposed to privately generate shots and instructions since either can leak information about the private training set. To overcome the problem that PATE-based approaches face with large output spaces (here equal to the vocabulary size of around 50k), DP-ICL [60] and DP-OPT [25] incorporate the EM and its improved versions [17, 21, 64] to privately release a token with the maximum count based on the voting from teacher prompts.

# 3 Prompt-based Private Adaptations for Text Generation

While PromptDPSGD and PromptPATE [15] were designed for classification tasks only, we further extend them to text generation tasks. Having prompt-based generation holds the advantage that, in contrast to fine-tuning based approaches, they support mixed-task inference [30, 33, 37], *i.e.,* they require one frozen model for multiple tasks rather than a separate model copy for each of them. This reduces storage and offers greater flexibility and efficiency.

**PromptDPSGDGen.** We observe that an adequate choice of hyperparameters is sufficient for adjusting PromptDPSGD [15] to generation tasks. This is in line with prior work highlighting that the challenge of prompt tuning is that it requires experimenting with various hyperparameter choices to achieve good performance [37]. In particular, we observe that increasing the number of parameters in the soft prompt from 0.1% of the total LLM parameters, as done for classification [15], to 10% of total model parameters, by enabling prefix projection, yields a significant increase in generation performance. Additionally, we observe the need for an increased learning rate, compared to other tuning methods, to generate more precise outputs. Otherwise, the hyperparameters are dependent on the data the model is trained on.

**PromptPATEGen.** Adjusting PromptPATE [15] to generation tasks (where more than one output token is generated) is challenging due to 1) the large output space (equivalent to the number of tokens in the vocabulary) and 2) the privacy costs incurred by generating multiple tokens through the teacher ensemble. To overcome this challenge and support generation tasks with an unlimited number of queries, we extended PromptPATE by combining the training of the student prompt from [15] with the privacy techniques used in [60] and call the result PromptPATEGen. In particular, PromptPATEGen uses the private generation in DP-ICL to obtain longer output sequences for some public data inputs. The outputs sequences can then be treated as a "label" for the public data and can be deployed as a form of student prompt, just like in PromptPATE [15].

# 4 Comparing Open and Closed LLM Adaptations

We perform a thorough conceptual and empirical study to compare the adaptation of both open LLMs with private tuning (PromptDPSGD [15], PrivateLoRA [61], and DP-FineTune [35]) and closed LLMs with private ICL (DP-ICL [60], PromptPATE [15], DP-FewShotGen [53], and DP-OPT [25]). Our comparison spans the axes of privacy protection, performance, and cost.

## 4.1 Comparing Privacy Protection

All the considered methods offer privacy guarantees according to DP. Thereby, they ensure that the final prompted LLM's predictions will not leak more than the specified tolerated privacy budget $\varepsilon$ to any party who queries the LLM or gets access to the final private prompt. Yet, the threat model of multiple private ICL methods for closed LLMs does not include providing privacy against the LLM provider. Those methods that do might still occasionally experience leakage. We analyze the result of this lack of consideration for the goal of truly privacy-preserving LLM adaptations. In our analysis, we distinguish between the leakage of private training data and the leakage of test data queried at inference time, which might also be sensitive.

**Private Training Data.** PromptPATE [15], DP-ICL [60], and DP-FewShotGen [53] (without using an open LLM) disclose (large parts of) their private training set to the LLM provider in the form of shots in their teacher prompts and their engineering. This leakage is inherent in their design. To avoid such leakage, DP-OPT [25] tunes the prompt locally with DP guarantees and then exposes it to the LLM provider. Thereby, the data that the prompt was generated from is protected towards the LLM provider with the DP guarantees that also protect against leakage to a querying party. While the experimental evaluation in [25] suggests that at higher $\varepsilon$, the locally generated DP prompts might still contain generated data close to the private training data, this is a step towards the right direction. However, to generate the private prompt, DP-OPT [25] requires a powerful open LLM deployed locally. Looking at Figure 1, it becomes obvious that any private tuning method executed on that open LLM would, conceptually, improve privacy protection since the LLM provider would neither be involved in the adaptation nor in the use of the adapted LLM, yielding absolute privacy against them.

**Private Query Data.** DP does not aim at protecting query data. Hence, none of the studied private ICL methods attempt to protect that data against the LLM provider. While the protection of query data is often considered as an orthogonal research direction, we note that all the private tuning-based adaptations of the open local LLMs do naturally prevent leakage of the query data to the LLM provider. This is because the querying party directly interacts with the data owner (see Figure 1)—making the use of open models inherently more suited for truly privacy-preserving application than relying on closed models.

## 4.2 Comparing Performance

We look at privacy-utility trade-offs to compare the performance of private tuning on open LLMs vs. private ICL on their closed counterparts. We depart from analyzing the trade-offs and required assumptions conceptually and then present our thorough experimental evaluation.

**Private Tuning Outperforms Private ICL Conceptually.** Previous work [37] has shown for the non-private settings that gradient based tuning methods (used for open LLMs) offer better accuracy and significantly lower computational costs than ICL (used for closed LLMs) since the adaptations can leverage the internal behavior of the LLM. This benefit holds also in the privacy regime. Moreover, the tuning based methods do not make additional assumptions, such as the availability of public data (required by PATE-based methods, such as PromptPATE [15]), making them inherently more practical.

**Private Tuning outperforms Private ICL Experimentally.** To assess the performance of private tuning vs. private ICL, we perform extensive experimental evaluation. We use various LLM architectures and multiple datasets for classification and text generation tasks.

### 4.2.1 Experimental Setup

**Text Classification.** We follow the setup from [25] and use four datasets for the evaluation: SST2 from the GLUE benchmark [59], Trec [34], Mpqa [42] and Disaster [5]. SST2 and Mpqa are two-class sentiment analysis datasets. SST2 includes 67.3k training samples and 872 test samples, while Mpqa contains 8.6k training samples and 2k test samples. Trec is a six-class question-type classification dataset with 5.4k training samples and 500 test samples. Finally, the Disaster dataset involves determining whether a sentence is relevant to a disaster scenario or not and includes 4.4k training and 1000 test samples.

**Text Generation.** We use three different datasets: SAMSum, a dialog summarization [22] (14732 train, 818 val, and 819 test samples), PFL-DocVQA, question answering [56] (85k train and 10k test samples), and MIT Movies trivia10k13, movie extraction on directors (MIT-D with 1561 train and 415 test samples) and genre (MIT-G with 2953 train and 780 test samples) [38].

**Closed Models.** We follow the setup and choice of models originally proposed in the respective previous papers to evaluate the four private ICL methods for closed LLMs [15, 25, 53, 60]. The GPT3-Babbage and GPT3-Davinci models cited in [53, 60] were discontinued in early 2024[2] and replaced by their second versions (babbage-002 and davinci-002). Therefore, we use the newer versions here. The (estimated) number of parameters for the closed models is: 1.3B for GPT3 Babbage, 175B for GPT3 Davinci, 1.76T for GPT4 Turbo, and 200B for Claude 2.1.

**Open Models.** We consider various open LLMs with differing pre-training sets and numbers of parameters to simulate the choices a data owner can make for their local LLM. We select the following models: Pythia [6], OpenLLaMA [20], Vicuna [10], Mixtral [28], Bart [31], and RoBERTA [41], whose sizes vary from 160M to 45B parameters.

### 4.2.2 Performance of Private Adaptations for Classification

We show that the private adaptations on local open LLMs outperform the private methods for closed LLMs for classification tasks. In Table 3, we analyze the performance differences. We follow the evaluation in [25] (Table 2) and average the accuracy across the tasks (denoted as Average). Our analysis follows the standard practice and sets the privacy budget as $\varepsilon = 8$ and $\delta = 1/|D|$ where $|D|$ is the training size [15, 25]. Among the methods for closed LLMs, DP-OPT was tested on

---

[2]https://platform.openai.com/docs/deprecations

Table 3: **Private local adaptations on open LLMs outperform their closed alternatives for classification tasks.** The default privacy budget is set to $\varepsilon = 8$, except for PromptPATE [15], where the performance plateaus after $\varepsilon = 0.3$. The best result for a given task is bolded, and the 2nd best is underlined. **T($)** is training cost while **Q($)** is query cost for 10k queries (SST2), **All($)** is total cost.

| Method | LLM Type | Model | SST2 | Trec | Mpqa | Disaster | Average | T($) | Q($) | All($) |
|---|---|---|---|---|---|---|---|---|---|---|
| 0-shot ($\varepsilon = 0$) [25] | Closed | GPT3 Davinci | $92.4_{\pm 0.0}$ | $51.8_{\pm 0.2}$ | $84.5_{\pm 0.1}$ | $76.4_{\pm 0.2}$ | 76.3 | 0 | 6.00 | 6.00 |
| DP-OPT (original) [25] | Closed | GPT3 Davinci | $92.2_{\pm 0.8}$ | $68.7_{\pm 6.5}$ | $85.8_{\pm 0.7}$ | $78.9_{\pm 0.3}$ | 81.4 | 2.10 | 6.00 | 8.10 |
| *ICL ($\varepsilon = \infty$) [25]* | *Closed* | *GPT3 Davinci* | $94.7_{\pm 0.4}$ | $79.1_{\pm 0.5}$ | $88.8_{\pm 0.1}$ | $69.0_{\pm 5.9}$ | 82.9 | 0 | 6.00 | 6.00 |
| PromptPATE [15]($\varepsilon = \infty$) | Closed | GPT3 Babbage | 93.8 | 58.7 | 83.0 | 64.3 | 75.0 | 8.66 | 1.72 | 10.38 |
| PromptPATE [15]($\varepsilon < 0.3$) | Closed | GPT3 Babbage | $88.8_{\pm 2.3}$ | $52.8_{\pm 1.5}$ | $79.0_{\pm 0.5}$ | $58.0_{\pm 0.5}$ | 69.6 | 9.72 | 1.72 | 11.44 |
| PromptPATE [15]($\varepsilon < 0.3$) | Closed | Claude 2.1 | $95.7_{\pm 1.4}$ | $79.3_{\pm 1.2}$ | $\mathbf{92.1_{\pm 0.6}}$ | $71.0_{\pm 0.8}$ | 84.5 | 48.24 | 5.36 | 53.6 |
| DP-FewShotGen(1) [53] | Closed | GPT3 Babbage | $72.8_{\pm 7.7}$ | $51.3_{\pm 5.8}$ | $73.4_{\pm 8.5}$ | $59.2_{\pm 2.5}$ | 64.2 | 0.86 | 1.10 | 1.96 |
| DP-ICL [60] | Closed | GPT3 Babbage | $92.8_{\pm 0.9}$ | $26.3_{\pm 5.6}$ | $80.6_{\pm 0.9}$ | $50.6_{\pm 1.1}$ | 62.6 | 0 | 17.2 | 17.2 |
| DP-ICL [60] | Closed | GPT4 Turbo | $95.9_{\pm 0.1}$ | $16.2_{\pm 1.7}$ | $90.4_{\pm 0.1}$ | $70.3_{\pm 0.4}$ | 68.2 | 0 | 138.00 | 138.00 |
| PromptDPSGD [15] | Open | RoBERTA Large | $92.3_{\pm 0.5}$ | $54.5_{\pm 2.5}$ | $50.0_{\pm 0.0}$ | $77.8_{\pm 0.6}$ | 68.6 | 7.59 | 0.40 | 7.99 |
| DP-FineTune [35] | Open | RoBERTA Large | $93.5_{\pm 0.3}$ | $93.7_{\pm 0.8}$ | $88.2_{\pm 0.4}$ | $\mathbf{82.2_{\pm 0.3}}$ | 89.4 | 5.75 | 0.40 | 6.15 |
| PrivateLoRA [61] | Open | RoBERTA Large | $93.6_{\pm 0.3}$ | $93.9_{\pm 0.6}$ | $87.7_{\pm 0.8}$ | $81.8_{\pm 0.2}$ | 89.3 | 3.45 | 0.40 | 3.85 |
| PrivateLoRA [61] | Open | Vicuna 7B | $94.8_{\pm 0.5}$ | $\mathbf{97.3_{\pm 0.1}}$ | $87.8_{\pm 0.5}$ | $81.3_{\pm 0.5}$ | **90.3** | 13.80 | 0.78 | 14.58 |
| DP-OPT (local) [25] | Open | Vicuna 7B | $89.5_{\pm 2.6}$ | $65.3_{\pm 4.3}$ | $80.7_{\pm 3.3}$ | $65.6_{\pm 0.3}$ | 75.3 | 2.10 | 0.78 | 2.88 |
| PrivateLoRA [61] | Open | Pythia 6.9B | $92.2_{\pm 0.5}$ | $96.3_{\pm 0.8}$ | $87.2_{\pm 0.3}$ | $82.1_{\pm 0.2}$ | 89.4 | 13.80 | 0.78 | 14.58 |
| PrivateLoRA [61] | Open | Pythia 160M | $80.4_{\pm 0.7}$ | $82.5_{\pm 3.2}$ | $77.9_{\pm 0.3}$ | $73.6_{\pm 0.2}$ | 78.6 | 1.60 | 0.50 | 2.1 |
| PrivateLoRA [61] | Open | Llama3-8B(Instruct) | $\mathbf{96.0_{\pm 0.1}}$ | $\underline{96.8_{\pm 0.2}}$ | $87.3_{\pm 0.2}$ | $80.8_{\pm 0.1}$ | $\underline{90.2}$ | 27.60 | 0.78 | 28.38 |

the strongest Davinci model (with 175B parameters) from the GPT3 family. Across all the tasks, DP-OPT is outperformed by both DP-FineTune and PrivateLoRA by a large margin (even >26% absolute on Trec), even though DP-FineTune and PrivateLoRA were trained on RoBERTa Large with only 355M parameters (500X fewer than for GPT3 Davinci). Furthermore, we show that PrivateLoRA outperforms DP-OPT even when using Pythia-6.9B, which guarantees that the open LLM for PrivateLoRA was not pre-trained on any of the downstream datasets. For a fair comparison, we also train PrivateLoRA on Vicuna 7B, which was used in DP-OPT as the local model to find the transferable prompts and show that PrivateLoRA is also significantly better than DP-OPT applied either directly to Vicuna 7B or when run on GPT3 Davinci. This suggests that the data owners, rather than using their local LLM to tune prompts for DP-OPT, should privately tune it with PrivateLoRA (in this case on RoBERTA Large) since it yields stronger performance and privacy at a lower cost.

For PromptPATE, the performance plateaus after around $\varepsilon = 0.3$, since it creates a public prompt using only a few shots, and the selection of the demonstrations from a large pool of publicly labeled examples has a negligible gain on the final performance. In the limit, we also show that PromptPATE even with an infinite privacy budget ($\varepsilon = \infty$) for GPT3 Babbage (with 1.3B parameters) performs worse than PrivateLoRA or DP-FineTune on RoBERTA Large (3.6X fewer parameters). In the same setup of models, PrivateLoRA and DP-FineTune on RoBERTA Large also outperform DP-ICL tested on GPT3 Babbage on all tasks. Additionally, PrivateLoRA adapted on Pythia-160M (with even fewer parameters) performs much better than DP-FewShotGen on GPT3-Babbage (8X more parameters).

We also run DP-ICL with GPT4 Turbo. The resulting accuracies are high for sentiment classification with SST-2 and Mpqa. However, it has the lowest accuracy on Trec (with 6 classes), caused by a small number of output probability tokens released for a query (only 20 vs 100 for GPT3, which might not contain the correct class label token) while being the most expensive option. Similar trends are observed for PromptPATE on Claude, however, it has more consistent performance and emerges as the most performant closed model on the tested tasks (while being the 2nd most expensive one). In contrast, Private LoRA with Vicuna 7B performs the best on Trec and on *average*. It is the best of all tested adaptations while incurring around 3.7 and 9.5 times lower costs than Claude and GPT4 Turbo, respectively. In general, the open models have the highest average performance at a much lower cost.

We further analyze the privacy-utility trade-off for classification tasks across different privacy budgets ($\varepsilon \in [0, 8]$) in Figure 2. We show that even under tight privacy constraints ($\varepsilon < 1.0$), the privacy-preserving adaptation for open LLMs performs significantly better than the one for closed LLMs. Specifically, we analyze the differences between PrivateLoRA for open LLMs vs PromptPATE for closed LLMs. The performance for PromptPATE plateaus after around $\varepsilon = 0.3$ and only for one out of four datasets, namely for MPQA, we observe that the crossover point between PromptPATE and PrivateLoRA (PromptPATE performs better than PrivateLoRA until $\varepsilon = 0.6$). For the smallest $\varepsilon = 0.1$ values that we analyzed, the performance of PrivateLoRA is better by 0.6% on SST2, by

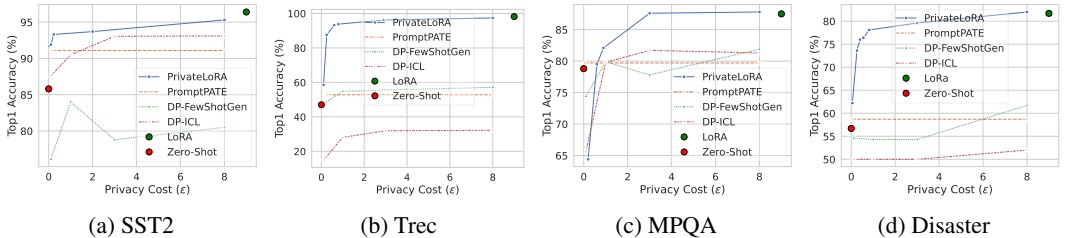

| (a) SST2 | (b) Trec | (c) MPQA | (d) Disaster |

Figure 2: **Privacy-utility trade-off for classifications tasks.** We use PrivateLoRA to adapt Vicuna-7b to the downstream tasks, PromptPATE, DP-ICL, and DP-FewShotGen with GPT3 Babbage. We analyze the privacy costs $\varepsilon$ in the range $[0, 8]$ (see corresponding Figure 3 for text generation tasks).

Table 4: **Evaluation on Dialog Summarization with SAMSum** for $\varepsilon = 8$. **T($)** is training cost while **Q($)** is query cost for 10k queries, **All($)** is total cost.

| Method | LLM Type | Model | Rouge-1 | Rouge-2 | Rouge-L | T($) | Q($) | All($) |
|---|---|---|---|---|---|---|---|---|
| DP-ICL [60] | Closed | GPT3 Davinci | $41.2_{\pm 0.6}$ | $16.3_{\pm 0.4}$ | $31.8_{\pm 0.3}$ | 0 | 665.91 | 665.91 |
| DP-ICL [60] | Closed | GPT3.5 Turbo | $42.6_{\pm 0.2}$ | $18.9_{\pm 0.3}$ | $33.8_{\pm 0.5}$ | 0 | 449.16 | 449.16 |
| DP-ICL [60] | Closed | GPT4 Turbo | $41.8_{\pm 0.2}$ | $17.3_{\pm 0.3}$ | $33.4_{\pm 0.2}$ | 0 | 3419.42 | 3419.42 |
| **PromptPATEGen** | Open | Vicuna 7B | $41.3_{\pm 0.3}$ | $18.0_{\pm 0.4}$ | $32.8_{\pm 0.2}$ | 3.29 | 2.74 | 6.03 |
| **PromptPATEGen** | Open | OpenLLaMA 13B | $43.4_{\pm 0.3}$ | $19.7_{\pm 0.5}$ | $34.2_{\pm 0.4}$ | 18.63 | 0.80 | 19.43 |
| **PromptDPSGDGen** | Open | BART-Large | $46.1_{\pm 0.4}$ | $21.3_{\pm 0.1}$ | $37.4_{\pm 0.0}$ | 1.73 | 0.40 | 2.13 |
| PrivateLoRA [61] | Open | BART-Large | $48.8_{\pm 0.6}$ | $23.5_{\pm 0.5}$ | $39.1_{\pm 0.2}$ | 2.90 | 0.69 | 3.59 |
| PrivateLoRA [61] | Open | Pythia 410M | $40.4_{\pm 0.1}$ | $16.6_{\pm 0.3}$ | $33.0_{\pm 0.4}$ | 3.45 | 1.34 | 4.79 |
| **PromptDPSGDGen** | Open | Pythia 1B | $41.2_{\pm 0.2}$ | $17.8_{\pm 0.1}$ | $33.7_{\pm 0.1}$ | 4.83 | 0.95 | 5.78 |
| DP-FineTune [35] | Open | Pythia 1B | $42.5_{\pm 0.7}$ | $18.4_{\pm 0.3}$ | $33.9_{\pm 0.3}$ | 9.84 | 1.08 | 10.92 |
| PrivateLoRA [61] | Open | Pythia 1B | $42.3_{\pm 0.6}$ | $18.4_{\pm 0.7}$ | $34.7_{\pm 0.5}$ | 4.24 | 1.00 | 5.24 |
| PrivateLoRA [61] | Open | Pythia 6.9B | $45.6_{\pm 0.3}$ | $21.4_{\pm 0.3}$ | $37.4_{\pm 0.5}$ | 10.18 | 6.57 | 16.75 |
| PrivateLoRA [61] | Open | Vicuna 7B | $48.6_{\pm 3.5}$ | $24.8_{\pm 2.6}$ | $40.2_{\pm 3.4}$ | 11.28 | 6.19 | 17.47 |
| PrivateLoRA [61] | Open | OpenLLaMA 13B | $48.5_{\pm 1.1}$ | $24.2_{\pm 0.8}$ | $40.1_{\pm 0.9}$ | 19.46 | 8.05 | 27.51 |
| PrivateLoRA [61] | Open | Mixtral 8x7B | $\mathbf{52.8}_{\pm 0.4}$ | $\mathbf{29.6}_{\pm 0.2}$ | $\mathbf{44.7}_{\pm 0.2}$ | 57.96 | 9.99 | 67.95 |

Table 5: **Evaluation on Question Answering with PFL-DocVQA** for $\varepsilon = 8$.

| Method | LLM Type | Model | Rouge-1 | BLEU | Levenshtein | T($) | Q($) | All($) |
|---|---|---|---|---|---|---|---|---|
| DP-ICL [60] | Open | OpenLLaMA 13B | $60.7_{\pm 0.6}$ | $23.9_{\pm 0.5}$ | $52.5_{\pm 1.1}$ | 0 | 641.32 | 641.32 |
| **PromptPATEGen** | Open | Vicuna 7B | $31.7_{\pm 1.5}$ | $26.6_{\pm 0.7}$ | $35.7_{\pm 0.4}$ | 2.28 | 0.57 | 2.85 |
| **PromptDPSGDGen** | Open | Pythia 1B | $57.3_{\pm 0.9}$ | $40.1_{\pm 1.1}$ | $66.8_{\pm 0.7}$ | 37.26 | 0.96 | 38.22 |
| DP-FineTune [35] | Open | Pythia 1B | $\mathbf{70.2}_{\pm 0.2}$ | $\mathbf{55.7}_{\pm 0.3}$ | $\mathbf{78.3}_{\pm 0.3}$ | 137.06 | 1.32 | 138.38 |
| PrivateLoRA [61] | Open | Pythia 1B | $64.2_{\pm 0.7}$ | $43.2_{\pm 0.8}$ | $73.4_{\pm 1.3}$ | 44.16 | 1.28 | 45.44 |
| PrivateLoRA [61] | Open | Pythia 6.9B | $64.4_{\pm 0.1}$ | $47.9_{\pm 0.2}$ | $73.3_{\pm 0.2}$ | 293.25 | 5.80 | 299.05 |
| PrivateLoRA [61] | Open | OpenLLaMA 13B | $63.1_{\pm 1.1}$ | $22.2_{\pm 1.3}$ | $70.7_{\pm 2.1}$ | 358.80 | 9.02 | 367.82 |

Table 6: **Evaluation on information extraction with MIT-D and MIT-G** for $\varepsilon = 8$.

| Method | LLM Type | Model | MIT-D | MIT-G | T($) | Q($) | All($) |
|---|---|---|---|---|---|---|---|
| DP-FewShotGen [53] | Closed | GPT3 Davinci | 80.6 | 64.1 | 0.42 | 2.36 | 2.78 |
| **PromptPATEGen** | Open | Vicuna 7B | $74.1_{\pm 0.6}$ | $41.7_{\pm 1.6}$ | 0.52 | 0.73 | 1.25 |
| **PromptPATEGen** | Open | OpenLLaMA 13B | $70.9_{\pm 0.5}$ | $33.4_{\pm 1.3}$ | 3.11 | 0.80 | 3.91 |
| PrivateLoRA [61] | Open | Pythia 410M | $74.3_{\pm 8.3}$ | $64.3_{\pm 2.8}$ | 0.06 | 0.50 | 0.56 |
| **PromptDPSGDGen** | Open | Pythia 1B | $89.8_{\pm 0.3}$ | $69.1_{\pm 1.7}$ | 0.17 | 0.25 | 0.42 |
| DP-FineTune [35] | Open | Pythia 1B | $92.2_{\pm 1.1}$ | $71.6_{\pm 1.1}$ | 0.94 | 0.50 | 1.44 |
| PrivateLoRA [61] | Open | Pythia 1B | $90.2_{\pm 0.1}$ | $68.8_{\pm 0.8}$ | 0.08 | 0.31 | 0.39 |
| PrivateLoRA [61] | Open | Vicuna 7B | $\mathbf{95.0}_{\pm 0.2}$ | $74.4_{\pm 1.2}$ | 0.52 | 5.92 | 6.44 |
| PrivateLoRA [61] | Open | OpenLLaMA 13B | $94.0_{\pm 0.8}$ | $\mathbf{76.4}_{\pm 0.9}$ | 1.04 | 6.21 | 7.25 |
| PrivateLoRA [61] | Open | Mixtral 8x7B | 93.0 | 69.7 | 1.52 | 9.47 | 10.99 |

4.4% on Trec, and by 3.5% on Disaster. Overall, the private adaptations for open LLMs outperform the ones for closed LLMs in most privacy regimes.

### 4.2.3 Performance of Private Adaptations for Text Generation

The evaluation of the three text generation tasks demonstrates superior performance of private adaptations on open vs closed LLMs. We consider the privacy-preserving ICL methods of DP-ICL and DP-FewShotGen on closed LLMs, since only these methods were executed for generative tasks. For the SAMSum datasets in Table 4, the first three adaptations (including our PromptPATEGen) are based on few-shot in-context learning (using discrete prompts), while the remaining results are for the private gradient-based adaptations. For the discrete prompts, our PromptPATEGen runs on local open Vicuna 7B and outperforms other discrete prompt-based methods from closed LLMs. Our PromptDPSGDGen performs on par with the other private tuning method (PrivateLoRA) run on Pythia 1B. Note that only PromptDPSGDGen and ICL adaptations (PromptPATEGen and DP-ICL) support multi-task inference.

We additionally leverage BART-Large (with 355M parameters) [1] that was fine-tuned on the XSum summarization task [45] (which does not include SAMSum). This specialized open model outperforms other LLMs apart from Vicuna with 7B parameters, OpenLLaMA with 13B parameters, and Mixtral with 45B parameters. Crucially, PrivateLoRA on BART-Large outperforms DP-ICL run on GPT3 Davinci, despite using the model with around 500X fewer parameters. This further indicates that we can leverage a large selection of open models to solve a specific task at lower cost and with better privacy protection without resorting to general-purpose closed LLMs. We also use PrivateLoRA on larger models from different families (Vicuna 7B, OpenLLama 13B, and Mixtral 8x7B) and observe that its performance and cost steadily increase with more parameters.

The evaluation on PFL-DocVQA in Table 5 shows that PrivateLoRA on open LLMs outperforms DP-ICL (which was run also only on OpenLLaMA 13B in the original paper [60] due to the cost constraints). We also evaluate both MIT-D and MIT-G in Table 6 on the accuracy of predicted vs target labels following the metrics in DP-FewShotGen. The adaptations of open LLMs with privacy-preserving gradient-based methods outperform DP-FewShotGen on the significantly larger GPT3 Davinci, for example, on MIT-D by 13.4% and on MIT-G by 22.3% absolute, respectively by PrivateLoRA on OpenLLaMA 13B.

We also present the privacy-utility trade-off for the SAMSum, MIT-G, and MIT-D datasets with varying values of $\varepsilon$ across the PrivateLoRA, PromptPATEGen, and DP-FewShotGen methods in Figure 3. We use the Pythia 1B model for MIT-D and MIT-G and the BART-Large model for SAMSum. The graphs clearly demonstrate a similar trend to that shown previously in Figure 2: PrivateLoRA for open LLMs significantly surpasses the performance of both DP-ICL and DP-FewShotGen, which rely on GPT-3 Davinci.

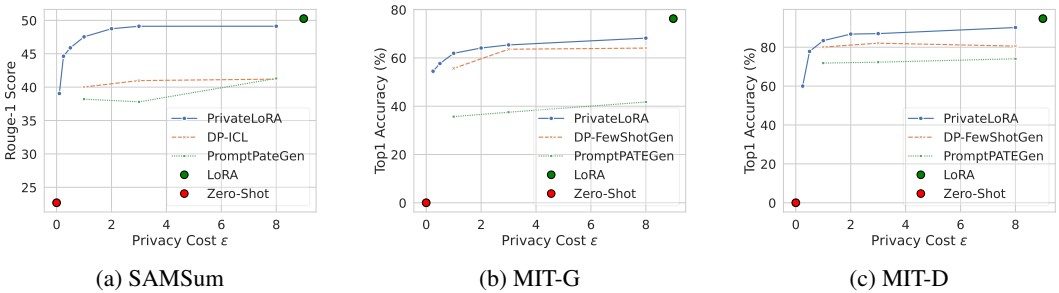

|  |  |  |
|:---:|:---:|:---:|
| (a) SAMSum | (b) MIT-G | (c) MIT-D |

Figure 3: **Privacy-utility trade-off for generation tasks.** We analyze the privacy costs $\varepsilon$ in the range $[0, 8]$ for the three generation tasks. PrivateLoRA for open LLMs substantially outperforms DP-ICL and DP-FewShotGen, which both utilize GPT3 Davinci. PrivateLoRA for MIT-D and MIT-G is trained on the Pythia 1B model, and for SAMSum on the BART-Large Model. PromptPATEGen uses Vicuna 7B.

## 4.3 Comparing Costs

We compare the costs of obtaining a private predictor for a given downstream task using open vs closed LLMs. We use the wall clock time to capture the running time of methods for local open LLMs, which we then translate to the monetary cost that would be incurred if we ran the method on cloud-based hardware. For the adaptations of closed LLMs, we count the number of tokens used in the queries and obtained outputs from the APIs. The pricing from cloud providers and OpenAI forms the basis for the cost estimations, and we show the selected values in Table 22 in the Appendix. Further details on how the costs were calculated for each private ICL methods are presented in Appendix D. Based on the estimated costs in Tables 1,3,4,5, and 6, the privacy-preserving methods for open LLMs require much lower costs (and perform better) than for closed LLMs in the considered scenarios. The costs for classification tasks are relatively low, especially for closed LLMs, since the tasks are simple and the number of tokens (particularly for outputs) is small. However, the costs increase substantially for generation tasks, especially for the closed LLMs, where DP-ICL is around 150X more expensive than PrivateLoRA for dialog summarization. While larger models often incur higher costs, they do not necessarily imply higher performance. For example, smaller models like RoBERTA Large for classification or BART-Large for dialog summarization can obtain one of the highest performances at the lowest prices.

# 5  Discussion and Future Work

In summary, our results highlight that from the perspective of providing truly privacy-preservation adaptations, open LLMs are strictly preferable over closed LLMs, since their adaptations are more private, more performant, and more cost-effective. Going beyond the concrete existing methods studied in this work [15, 25, 53, 60], in the following, we analyze the general reasons behind the underwhelming results of privacy-preserving closed LLM adaptations.

**Privacy Leakage.** The enhanced privacy protection from adapting open LLMs is a major benefit: users' private training data and queries to adapted open LLMs are never revealed to third parties. On the contrary, the leakage of private query data to the LLM provider is to date an inherent problem with closed LLMs, since no methods to provide formal guarantees for the query data are currently known. Potential solutions might involve private inference for LLMs, where a model performs inference on encrypted queries, however, it is still in its nascency [9, 23, 32] for the scale of closed LLMs [7].

**Performance.** We argue that the lower performance of closed LLM adaptations stems from the fact that they have to rely on discrete prompts and that engineering such prompts for the closed LLMs is highly challenging. This is because 1) prompts, in general, have been shown to exhibit an unstable performance and to require a large number of trials and errors or discrete optimization while still underperforming gradient-based approaches [37]. Additionally, 2) when the prompts (for privacy reasons) are not tuned on the closed LLM but on an open LLM surrogate model, additional performance decrease is incurred through the prompt transfer, since it has been shown that transferred prompts cannot reach the performance of prompts directly tuned on a given LLM [52]. While the latter problem might be mitigated through the design of more performant prompt transfer techniques, the former one seems to be a more fundamental limitation [37].

**Costs.** The high costs incurred by some closed LLM adaptations result from the fact that they rely on ensemble-based approaches to yield DP guarantees and the fact that they incur continuous query costs at inference time. The former one could be solvable by designing more efficient DP schemes for discrete prompts, however, the latter is inherent to the nature of closed LLMs.

We hope that implementing the above-mentioned solutions will shrink the gap between private adaptations of open and closed LLMs. However, it remains unclear whether it is worth the community's effort, given the effectiveness of private adaptations for open LLMs.

## Broader Impacts

Our comparative study of open and closed LLMs has significant implications for private adaptations: our research advocates for the use of open LLMs for private adaptations.

We stress that our goal is not to discredit closed LLMs, but to highlight the potential privacy and performance benefits as well as cost-effectiveness associated with the use of open LLMs. Through thorough evaluations, we demonstrated in our paper that adapting open LLMs with private parameter efficient fine-tuning methods results in higher performance and mitigates open privacy risks of in-context learning with closed models. This not only leads to better performance but also reduces costs, making privacy adaptation on open LLMs a more viable option for many applications.

Moreover, our work can serve as a baseline for future private learning methods for LLMs. We believe that an open dialogue about the strengths and weaknesses of both open and closed LLMs is crucial for the advancement of privacy-preserving LLMs. We hope that our research will serve as a catalyst for further investigations into private adaptations of LLMs, ultimately leading to the development of models that effectively balance the need for both openness and privacy, all while ensuring that user privacy remains uncompromised.

### Acknowledgments

The project on which this paper is based was funded by the German Federal Ministry of Education and Research (BMBF) under funding number 16KIS2114K. This work was also supported by the German Research Foundation (DFG) within the framework of the Weave Programme under the project titled "Protecting Creativity: On the Way to Safe Generative Models" with number 545047250. Responsibility for the content of this publication lies with the authors.

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

## A  Limitations

**Hyperparameter tuning.**  Due to limited computational resources, we were not able to tune the hyperparameters for all open LLM private tuning methods. This is especially true for the generation tasks with longer input length, PFL-DocVQA for example, due to their long training time. Therefore, we still see the potential for an increase in performance. But, as our results for private tuning already show that these methods outperform the private in-context learning approaches, the used hyperparameters suffice for these comparisons.

**Pretraining set as privacy risk.**  Closed models do not provide much information about their pretraining process. This also includes the non-disclosure of the pretraining set. For this reason, we purposefully chose open LLMs where the pretraining set is known and can be easily downloaded. But, even though the pretraining set is available, and a user could check it directly against their own data for any privacy leakage, we recognize that this process is also costly. This is why, despite having potential access to the pretraining set, a user might not be able to cover the privacy risk fully.

**Cost estimation.**  To show the potential cost of running the training of privately tuning an open LLM, we used the cost of $0.69 per compute hour from RunPod. This represents the average cost of running such training, and the cost might vary for different users on machines of different cloud providers. Additionally, we ran our training on our own machines, therefore, do not have the exact same server setup as the one given by RunPod. This could lead to additional variations in training time and therefore training costs.

**Budgetary limits.**  In this paper we looked at different closed models (GPT3 Davinci, GPT3 Babbage, GPT4 Turbo, and Claude 2.1) and methods to add private in-context learning. Due to limitations to our budget, we were not able to look at a higher variety of closed models and also extend the experiments with GPT4 Turbo.

**Limited model sizes.** In our experiments for the private tuning of open LLMs, we tried a variety of different models and model sizes. To handle the models with a higher amount of parameters, we needed to use 4-bit loading to be able to run the training on our GPUs. Unfortunately, due to some type-inconsistencies with Opacus, the Python library we use to run DP-SGD, and 4-bit loading from bitsandbytes, we were not able to do training on our machines for some of the models. This included 13B models for PrivateLoRA for the classification tasks and 7B+ models for PromptDPSGDGen and DP-FineTune for the text generation tasks.

# B    Further Details on the Related Work

## B.1    Private Adaptations for Closed LLMs

We present the methods for the private adaptations of closed LLMs with a few more details.

**PromptPATE [15]** prompts an LLM with different prompts containing disjoint examples from the private training dataset, each prompt corresponding to a teacher. To label the public data for the knowledge transfer, PromptPATE [15] infers the next token prediction of each teacher on public text sequences and interprets them as labels. Instead of training a student model from scratch, PromptPATE [15] creates a student prompt. It utilizes the data efficiency of discrete prompts by selecting examples for the student prompt from the labeled public sequences.

**DP-ICL** [60]. For the generation tasks, it proposes the Embedding Space Aggregation(ESA), which involves mapping each sentence produced by the LLM for a given exemplar-query pair onto the embedding space and then reconstructing a sentence from the noisy mean of these embeddings. This process depends on the quality of the text-to-embedding models and the zero-shot examples employed to map the noisy mean embedding back to the sentence, potentially leading to suboptimal outputs. The other approach proposed in DP-ICL is keyword space aggregation (KSA). It creates a *keyword space* by segmenting each output sentence into keywords to form a histogram. The keywords with the highest counts are selected privately using either the Propose-Test-Release (PTR) or Joint Exponential Mechanism (JointEM) [21]. The selected private keywords are then used to create a prompt and query the LLM.

**DP-FewShotGen** [53] introduces a method for text generation of public prompts. In this method, tokens are individually generated using disjoint subsets of the private data and then noisily aggregated based on the frequency of the generated tokens to predict the next token. The drawback of this approach is that the generation process is conditioned on the label. Consequently, despite being a text generation task, it necessitates the assignment of a public label to the private data.

**DP-OPT** [25] is currently the only private ICL method that uses discrete prompt tranferrability to create a private prompt on a local open model, which can be used to infer a closed model. Based on the approach of deep language networks [51], multiple initial prompts with different private examples are optimized through separate back- and forward passes such that a prompt is created that gives good performance on the downstream task. To add privacy, they use the exponential mechanism to sample each generated token from all different initial prompts. Currently, their proposed method is only shown to work with classification tasks.

## B.2    Private Text Generation based on PATE

**SeqPATE** [55] safeguards the privacy of individual training samples and sensitive phrases in the training data of a language model. To adapt PATE for text generation, SeqPATE creates pseudo-contexts, simplifying the sequence generation task to a next-word prediction problem. To manage the extensive output space, SeqPATE introduces a candidate filtering strategy that dynamically narrows the output space and enhances the teacher aggregation in PATE to avoid low agreement caused by voting among a large number of candidates. Additionally, to further minimize privacy losses, it employs knowledge distillation to reduce the number of teacher queries.

# C    Additional Details on our Setup

In this section, we present the detailed (hyper-)parameters used to evaluate all the tasks that were used for the different Open and Closed LLMs privacy-preserving training methods.

## C.1 Text classification

**Detailed information about the datasets.** We expose the different statistics of each dataset used for text classification evaluation in Table 7. For SST2, the validation set was used as the test set, as the original test set is only provided with unknown labels for each sample.

Table 7: **Stastistics of the 4 evaluated tasks** related to text classification.

| Task | #Train | #Test | #Class | Task description |
|---|---|---|---|---|
| SST2 | 66,674 | 872 | 2 | Sentiment analysis on movie reviews |
| Trec | 5,452 | 500 | 6 | Question type classification |
| Mpqa | 8,603 | 2,000 | 2 | Sentiment analysis on short ensembles |
| Disaster | 4,430 | 1,000 | 2 | Relevance of sentence to a disaster |

**Private Tuning.** We detail the hyperparameters used to fine-tune the models with private LoRA in Table 8, for DP-FineTune in Table 9 and for PromptDPSGD in Table 10. All the experiments were conducted on 3 different seeds. Note that unlike LoRA or Full-Finetune, PromptDPSGD requires a precise tuning of hyperparameters. A total of 50 trials over 100 epochs were necessary to tuned them. For the Mpqa sentiment analysis task, no converging set of hyperparameters was found.

Table 8: **Hyperparameters for PrivateLoRA [61]** on evaluated classification datasets for $\varepsilon = 8$.

| Hyperparameters | Datasets | | | |
|---|---|---|---|---|
| | SST2 | Trec | Mpqa | Disaster |
| bs | 128 | 128 | 128 | 128 |
| lr | 1e-3 | 1e-3 | 1e-3 | 1e-3 |
| max grad clip | 0.1 | 0.1 | 0.1 | 0.1 |
| epochs | 10 | 40 | 20 | 20 |
| lora rank | 4 | 4 | 4 | 4 |
| $\delta$ | $\frac{1}{|D|}$ | $\frac{1}{|D|}$ | $\frac{1}{|D|}$ | $\frac{1}{|D|}$ |
| GradClip | 0.1 | 0.1 | 0.1 | 0.1 |

Table 9: **Hyperparameters for DP-FineTune [35]** on evaluated classification tasks with Roberta-Large for $\varepsilon = 8$.

| Hyperparameters | SST2 | Trec | Mpqa | Disaster |
|---|---|---|---|---|
| LR | 1e-4 | 1e-4 | 1e-4 | 1e-4 |
| BS | 128 | 128 | 128 | 128 |
| Epoch | 10 | 40 | 40 | 50 |
| $\delta$ | $\frac{1}{|D|}$ | $\frac{1}{|D|}$ | $\frac{1}{|D|}$ | $\frac{1}{|D|}$ |
| GradClip | 0.1 | 0.1 | 0.1 | 0.1 |

Table 10: **Hyperparameters for PromptDPSGD [15].** The hyperparameters for SST2 datasets are directly extracted from the paper and are evaluated on Roberta-Large for $\varepsilon = 8$. LR = learning rate, BS = batch size, GRAD = per sample gradient clipping. P-length = length of the prepended prompt in number of tokens. The trainings are all performed with prefix-tuning and not soft-prompt. Those are the hyperparameters of the best performing prompt on the test set of each dataset, and the accuracy of this prompt is reported in the table.

| Hyperparameters | SST2 | Trec | Mpqa | Disaster |
|---|---|---|---|---|
| LR | 0.01 | 0.001 | - | 0.01 |
| BS | 32 | 32 | 32 | 32 |
| GRAD | 4 | 0.3 | - | 1.0 |
| Epochs | 22 | 100 | 100 | 100 |
| P-length | 1 | 10 | 10 | 10 |
| Best accuracy | 92.8 | 58.0 | 50.0 | 78.6 |

**Private in-context learning.** The respective set of hyperparameters for DP-FewShotGen, Prompt-PATE and DP-ICL are listed in Table 11, Table 12 and Table 13. For the used hyperparameters for DP-OPT, see [25] since the results of Table 3 are directly extracted from the paper. The accuracy results for DP-FewShotGen were computed for 5 different generated prompts following the method from the paper. For the PromptPATE method, experiments were only conducted for MPQA and Disaster datasets as we used already made evaluation from the original paper PromptPATE [15] for SST2 and Trec datasets on using GPT3-Babbage. All hyperparameters here are extracted directly from the previous paper.

Table 11: **Hyperparameters for DP-FewShotGen [53]** for the evaluation of new datasets with $\varepsilon = 8$ on GPT3-Babbage. M = Number of private prompts used for meta prompt generation. N = number of private shots per prompt. $\sigma$ = noise relative to wanted $\varepsilon$ using the Gumbel mecanism. $T_{max}$ = max number of tokens of the generate prompt.

| Hyperparameters | SST2 | Trec | Mpqa | Disaster |
|---|---|---|---|---|
| $\sigma$ ($\varepsilon = [0.1, 1, 3, 8]$) | [1.0,0.61, 0.48,0.34] | [3.0,0.83, 0.59,0.44] | [2.0,0.77, 0.57,0.41] | [3.5,0.93, 0.64,0.46] |
| MN | 80 | 80 | 80 | 80 |
| M | 20 | 20 | 20 | 20 |
| $T_{max}$ | 50 | 50 | 50 | 50 |

Table 12: **Hyperparameters for PromptPATE [15]** for the evaluation of new datasets with $\varepsilon = 8$ on GPT3-Babbage.Those parameters are common to all 4 tasks.

| Hyperparameters | Claude | GPT3-babbage |
|---|---|---|
| train set | 400 | 400 |
| student set | 200 | 300 |
| num shots | 2 | 1 |

Table 13: **Hyperparameters for DP-ICL [60]** for the evaluation of the text classification datasets with $\varepsilon = 8$ on GPT3-Babbage.

| Hyperparameters | SST2 | Trec | Mpqa | Disaster |
|:---:|:---:|:---:|:---:|:---:|
| num shots | 4 | 4 | 4 | 4 |
| Ensemble | 10 | 10 | 10 | 10 |
| Queries | 872 | 500 | 1000 | 1000 |

## C.2 Text Generation

We analyze the following generative downstream tasks: SAMSum, PFL-DocVQA, and MIT Movies trivia10k13. As we did for classification tasks, we compare the methods on closed LLMs against PrivateLoRA [61], PromptDPSGD [15], and DP-FineTune [35] that are run on open LLMs. For the PrivateLoRA [61] training, we use 4-bit quantization with QLoRA [14] to reduce the occupied GPU memory, which was implemented for the adaptations of open LLMs with more than 1B parameters on PFL-DocVQA and SAMSum datasets due to their long input sequences.

**Detailed information about the datasets.** We show the amount of data that we utilized in the experiments in Table 14.

Table 14: **Overview of the 4 text ge tasks** related to text generation.

| Task | #Train | #Test | Task description |
|:---:|:---:|:---:|:---:|
| SAMSum | 14,732 | 819 | Dialogue summarization |
| PFL-DocVQA | 85,000 | 10,000 | Question and answering |
| MIT-G | 2,953 | 780 | Extracting genres from movie reviews |
| MIT-D | 1,561 | 415 | Extracting directors from movie reviews |

**Private Tuning.** In Table 15, Table 16, and Table 17, we show the hyperparameters we used to train the open models with PrivateLoRA, PromptDPSGDGen, DP-FineTune respectively. For PrivateLoRA, we were able to use the same hyperparameters for all models for each task. In the tables, the *Max Seq Length* refers to the maximum amount of tokens of the sequence the model trains on. For *Schedulers*, we chose two different options, a constant scheduler that does not change the learning rate during training, and a linear scheduler. The linear scheduler is the default scheduler of the Hugging Face implementation of the Trainer class. It linearly decreases the learning rate over the whole training. For PromptDPSGDGen, we additionally have *Prefix Projection*. If enabled, prefix projection adds two additional linear layers to the prefix encoder. This increases the amount of trainable parameters, which in turn also increases the capability of the prefix to represent tasks. The evaluations for MIT-D, MIT-G, and SAMSum were done for 3 different seeds, whereas we used 2 different seeds for PFL-DocVQA.

Table 15: **Hyperparameters for PrivateLoRA [61]** on evaluated generation tasks for $\varepsilon = 8$. The hyperparameters are the same for the used models. The tested schedulers for MIT-G and MIT-D does not make a difference during training

| Hyperparameters | SAMSum | PFL-DocVQA | MIT-G | MIT-D |
|---|---|---|---|---|
| LR | 8e-4 | 8e-4 | 8e-4 | 8e-4 |
| BS | 256 | 256 | 256 | 256 |
| LoRA Rank | 8 | 8 | 8 | 8 |
| Max Seq Length | 650 | 1500 | 128 | 128 |
| Epoch | 20 | 15 | 20 | 20 |
| Scheduler | Linear | Linear | / | / |
| $\delta$ | $\frac{1}{|D|}$ | $\frac{1}{|D|}$ | $\frac{1}{|D|}$ | $\frac{1}{|D|}$ |
| GradClip | 0.1 | 0.1 | 0.1 | 0.1 |

Table 16: **Hyperparameters for PromptDPSGDGen** on evaluated generation tasks for $\varepsilon = 8$. The hyperparameters are the same for the used models. The tested schedulers for MIT-G and MIT-D do not result in difference in performance.

| Hyperparameters | SAMSum | PFL-DocVQA | MIT-G | MIT-D |
|---|---|---|---|---|
| LR | 1e-3 | 1e-3 | 1e-3 | 3e-3 |
| BS | 256 | 256 | 256 | 256 |
| P-Length | 10 | 25 | 5 | 5 |
| Prefix Projection | True | True | True | True |
| Max Seq Length | 650 | 1500 | 128 | 128 |
| Epoch | 20 | 15 | 40 | 40 |
| Scheduler | Linear | Linear | / | / |
| $\delta$ | $\frac{1}{|D|}$ | $\frac{1}{|D|}$ | $\frac{1}{|D|}$ | $\frac{1}{|D|}$ |
| GradClip | 0.1 | 0.1 | 0.1 | 1 |

Table 17: **Hyperparameters for DP-FineTune [35]** on evaluated generation tasks for $\varepsilon = 8$.

| Hyperparameters | SAMSum | PFL-DocVQA | MIT-G | MIT-D |
|---|---|---|---|---|
| LR | 8e-4 | 2e-4 | 2e-4 | 2e-4 |
| BS | 256 | 256 | 256 | 256 |
| Max Seq Length | 650 | 1500 | 128 | 128 |
| Epoch | 20 | 15 | 20 | 20 |
| Scheduler | Linear | Linear | Constant | Linear |
| $\delta$ | $\frac{1}{|D|}$ | $\frac{1}{|D|}$ | $\frac{1}{|D|}$ | $\frac{1}{|D|}$ |
| GradClip | 0.1 | 0.1 | 0.1 | 0.1 |

**Privacy-preserving prompt tuning.** In the following, we provide the used hyperparameters for the methods for Private ICL for Closed LLMs. In detail, for DP-FewShotGen in Table 18, for DP-ICL in Table 19, and for PromptPATEGen in Table 20.

Table 18: **Hyperparameters for DP-FewShotGen [53]** on evaluated generation tasks for $\varepsilon = 8$. We used the hyperparameters given in the original paper for MIT-G and MIT-D.

| Hyperparameters | SAMSum | MIT-G | MIT-D |
|:---:|:---:|:---:|:---:|
| $\sigma$ | 0.384 | 0.5 | 0.58 |
| MN | 80 | 80 | 80 |
| M | 20 | 20 | 20 |
| $T_{max}$ | 50 | 20 | 20 |

Table 19: **Original hyperparameters for DP-ICL [60]** on evaluated generation tasks for $\varepsilon = 8$.

| Hyperparameters | SAMSum | PFL-DocVQA |
|:---:|:---:|:---:|
| Model | GPT-Davinci | OpenLLaMA 13B |
| Ensemble | 100 | 100 |
| #Queries | 10,000 | 10,000 |

Table 20: **Hyperparameters for PromptPATEGen** on generation tasks for $\varepsilon = 8$.

| Hyperparameters | SAMSum | MIT-G | MIT-D |
|:---:|:---:|:---:|:---:|
| Model | Vicuna 7B | Vicuna 7B | Vicuna 7B |
| Ensemble | 100 | 25 | 25 |
| #Queries | 100 | 100 | 100 |
| #Student Prompt | 10 | 4 | 4 |
| $\sigma$ | 1.15 | 0.9 | 0.9 |

# D  Cost Calculation

We provide the details on measuring the cost for different methods. The assumed costs for interacting with the model APIs per 1 million tokens and GPU cost per hour are shown in Table 22. For the open LLMs, we set the median pricing per hour (based on prices from three GPU cloud providers shown in Table 22) which is $0.69 using an A40 GPU with 48GB of memory [3], which is a popular graphics card, also used in the previous work [15]. We note that we do refrain from using other metrics than monetary cost. For example, FLOPS are not a direct measurement of real-world computational cost because latency, power usage, and other costs can vary significantly depending on hardware and other factors [13].

**Costs for private-tuning-based adaptations.**   The private tuning-based adaptations of open LLMs require us to adjust the model parameters or the inputs for a given task, thus, we measure the running time of the training process and then query answering.

**Costs for private ICL-based adaptations.**   DP-ICL does not incur any training cost but uses an ensemble of teachers for each query (the same as PromptPATE for labeling public examples), which elevates the cost by the number of teachers, which can be 10 or even 100. For PromptPATE, the generation of public student prompts is done using an ensemble of teacher prompts, thus labeling each public data point costs much more (proportional to the number of teachers) than running a query (with a single prompt). DP-FewShotGen also uses an ensemble of prompts, where the number of accesses to the API in the training process is equal to the number of tokens in a public prompt. The cost of training the public prompt for DP-OPT is through the iterative process of instructing the local model to improve the prompt and obtain better predictions, however, this part is done on a local open

---

[3]The pricing is for the RunPod Cloud Service: `https://www.runpod.io/gpu-instance/pricing`.

LLM, thus, the cost is relatively low. For ease of approximation and to the benefit of the ICL methods, we assume that the creation of the teacher prompts and the private aggregation of the outputs have negligible costs. After preparing the public prompt, PromptPATE, DP-FewShotGen, and DP-OPT, need a single access to the API to answer a query.

To obtain the cost for closed LLMs, we have to compute the average number of tokens per query. For the classification task, we can take the example of the DP-OPT method applied on the SST2 dataset. For this dataset, only one token is returned by the API provider, so the cost of the outputs is negligible. SST-2 inputs have an average length of 12.35 and the best performing prepended prompt from DP-OPT training has a length of 39 tokens. Thus, for the DP-OPT task, for each query to the API, 41.35 tokens are sent approximately. This gives a cost of \$0.0006 per query for GPT-3 Davinci and the total cost of \$6 for 10k queries in Table 1. The cost per query is computed similarly, depending on the size of the prepended prompt of each ICL method. Regarding the generation task, we can take the example of the SAMSum dialog summarization dataset, in which the average token length is 141 for the input and 26 for the output, hence, a single query costs \$0.000333 (for GPT3-Davinci). The cost for a 0-shot inference to Davinci would therefore be \$3.33 for 10k queries. As DP-ICL considers the 1-shot scenario and an ensemble of 100 teachers, we add the average input and label lengths to the input and multiply this by the size of the ensemble, which results in an overall cost of roughly \$666. The exact average token count for each dataset which we used for the cost estimations can be found in Table 21.

Table 21: **Average token length** of different inputs and outputs of the used datasets. The average does not include instructions.

| Dataset | SST-2 | Trec | Mpqa | Disaster | MIT-D | MIT-G | SAMSum | DocVQA |
|---------|-------|-------|------|----------|--------|--------|---------|---------|
| Input | 12.35 | 11.43 | 3.88 | 30.79 | 25.276 | 24.314 | 140.857 | 924.191 |
| Output | 1 | 1 | 1 | 1 | 3.877 | 2.301 | 25.620 | 6.384 |

Table 22: **Pricing** for the models and cloud options (as of May 22nd, 2024).

| Model | Cost/1M tokens | | Cost/hour |
|-------|------|--------|-----------|
| | Input | Output | |
| GPT-Babbage [4] | \$0.40 | \$0.40 | - |
| GPT-Davinci | \$2.00 | \$2.00 | - |
| GPT-3.5-turbo Instruct | \$1.50 | \$2.00 | - |
| GPT-4-turbo | \$10.00 | \$30.00 | - |
| Claude 2.1 [5] | \$8 | \$24 | - |
| A40 (RunPod) [6] | - | | \$0.69 |
| A40 (Replicate) [7] | - | | \$2.07 |
| A40 (Hyperstack) [8] | - | | (starts from)\$0.50 |

# E  Additional Experiments

**PrivateLoRA extensive classification results.** Table 23 shows the top1 accuracies at different $\varepsilon$ used to compute the PrivateLoRA graph for each of the 4 text classification tasks in Figure 2.

---

[4]https://openai.com/api/pricing/

[5]https://www.anthropic.com/api

[6]https://www.runpod.io/gpu-instance/pricing

[7]https://replicate.com/pricing

[8]https://www.hyperstack.cloud/gpu-pricing

Table 23: **Private LoRA [61] top1-accuracies** for the evaluated datasets given different $\varepsilon$.

| $\varepsilon$ | Model | SST-2 | Trec | Mpqa | Disaster |
|---|---|---|---|---|---|
| 8 | Vicuna 7B | 95.3 | 97.4 | 88.4 | 82.0 |
| 3 | Vicuna 7B | 94.4 | 96.2 | 87.6 | 79.6 |
| 1 | Vicuna 7B | 93.5 | 93.8 | 82.1 | 78.1 |
| 0.7 | Vicuna 7B | 93.4 | 93.2 | 79.5 | 76.4 |
| 0.3 | Vicuna 7B | 91.9 | 87.6 | 64.4 | 73.6 |

**Safety Evaluation of Mixtral-8x7B Instruct with and without differential privacy.** We conducted additional experiments to analyze how fine-tuning a downstream task with and without differential privacy affects the safety alignment of models. We followed the approach from [63] to evaluate Mixtral-8x7B-instruct and fine-tuned the model on SAMSum once with $\varepsilon = 8$ and once with $\varepsilon = \infty$. We selected SAMSum as it does contain conversations with unsafe language (e.g., cursing or harassment). These results are presented in Table 24. The table is divided into two sections, "Compliance on Harmful Queries" and "Refusal on Harmless Queries". The scores were generated by separately prompting our model with 100 harmful and 100 harmless queries, each repeated 20 times with different safety prompts. Finally, the outputs are categorized as complying or refusing the input by Llama Guard, giving the percentage of incorrectly handled answers in our table. Therefore, the lower the score, the better.

First, we observe that fine-tuning of any kind decreases the model's safety capability. This is easily identifiable in the table, as the original model has lower scores across the board for compliance with harmful behavior compared to both fine-tuned models. Similar results were also shown in [29] and [62], where the authors used a dataset containing unsafe samples to fine-tune safety-aligned models, which drastically increases compliance with unsafe behavior.

Second, we can observe that private fine-tuning affects the safeguards less than non-private fine-tuning. In the non-private case, the influence of individual samples is unrestricted, which can amplify the impact of unsafe samples, as shown in [29] and [62]. In contrast, differential privacy limits the influence of any single sample. Consequently, the impact of unsafe examples is minimized, resulting in a model that retains more of its safe behavior.

Table 24: **Safety Evaluations of Mixtral-8x7B Instruct** Evaluating the safety of the responses given by different models. We compare the base instruction fine-tuned model with the same fine-tuned on SAMSum with and without differential privacy. The model responses were categorized as harmful and harmless by Llama Guard.

| Model | % Compliance on Harmful Queries ↓ | | | | % Refusal on Harmless Queries ↓ | | | |
|---|---|---|---|---|---|---|---|---|
| Mixtral-8x7B Instruct | short | Mistral | Llama | no prompt | short | Mistral | Llama | no prompt |
| Base model | 1 | 0 | 0 | 36 | 1 | 0 | 2 | 0 |
| $\varepsilon = \infty$ | 38 | 30 | 39 | 64 | 2 | 2 | 3 | 0 |
| $\varepsilon = 8$ | 9 | 5 | 7 | 48 | 2 | 0 | 1 | 0 |

**Analysis of PromptPATEGen and DP-ICL [60] across different generation tasks.** We observe that PromptPATEGen is particularly adept at summarization tasks, as shown in Table 4. However, when we shift focus to other task generation tasks, the performance of PromptPATEGen is not as impressive compared to DP-ICL [60]. For the MIT datasets, which are entity retrieval tasks, PromptPATEGen generates results that are semantically similar to the true values, but not identical. For instance, on the MIT-G dataset, PromptPATEGen generates "sci-fi" whereas the ground truth movie genre is "sci fi". This pattern of generating semantically similar results is also observed in the MIT-D dataset, where only one name is given as the movie director rather than the full names.

We further added evaluations using DP-ICL [60] with newer OpenAI models, such as GPT4 Turbo and GPT3.5 Turbo, for SAMSum and saw that the performance only increases slightly compared

Table 25: The dialogue, ground truth, and summary generated with **DP-ICL [60] on GPT4 Turbo** for a **SAMSum** sample

| | |
|---|---|
| Dialogue | Ursula: Haha I got a 93 on my French exam
Bob: Well done girl!
Jana: Wow
Jana: How did u manage to do that
Ursula: I just studied hard for it
Jana: omg
Jana: French is so hard
Vaughn: I got a 65
Vaughn: I didn't study for it haha
Ursula: At least you passed
Vaughn: Congrats! |
| Ground truth | Ursula got 93 on her French exam. Vaughn got 65, but still passed. |
| Generated summary | Ursula shares that she got a 93 on her French exam and Jana asks how she managed to do so well. Ursula says she studied hard for it. Vaughn admits to getting a 65 without studying and Ursula congratulates him for passing. |

to the original evaluations on GPT3 Davinci (see Table 4). Therefore, we looked through the summarizations that were given by GPT4 Turbo and compared them to the ground truths. Similarly to the above-mentioned problem with the MIT-G and MIT-D datasets, we found, that the results were semantically correct, but not close enough to the ground truth. As an example, we show a sample generation with the corresponding dialogue and ground truth in Table 25. We can see, that the generated summary is factually correct, but too extensive compared to the ground truth. In particular, this example results in a Rouge-1 score of 31.8. Our assumption, as we get more extensive, and thus, worse samples, is the limited examples in terms of summary structure given by the keywords provided by DP-ICL [60].

The DocVQA dataset presents a unique challenge for PromptPATEGen. Given the high average token length in this dataset, only a single example can be used as the in-context example. This limitation poses a significant challenge, as a single example may not adequately represent the diverse aspects present in the dataset.

**DP-FewShotGen [53] limitations on new GPT models.** In addition to running DP-FewShotGen [53] on GPT Babbage and GPT Davinci, we also adapted the official code to work with newer OpenAI models, such as GPT4. The code had to be changed, as OpenAI moved from their Completion API to the new Chat Completion API. During execution, we encountered an issue that prevents us from running experiments with the Chat Completions API with DP-FewShotGen [53]. The issue arises from the parallel use of logit bias and log-probabilities, as the resulting output while using both together is unusable for further processing. We assume, this behavior is caused by a defense mechanism against a new potential model-stealing attack [8], as one of the suggested defenses was to prohibit the use of both logit bias and log-probabilities at the same time, which would explain what we have experienced. As the implementation of DP-FewShotGen [53] uses both at the same time, we are unable to get this private ICL method to work with GPT models that use the Chat Completions API.

# F   Generation Metrics

In this section, we briefly discuss the different metrics we use to evaluate the generation tasks.

**Rouge [36].** The metrics in the Rouge, short for Recall-Oriented Understudy for Gisting Evaluation, set describe how many word-wise n-grams match between the predicted and target text. For Rouge-1, we look at uni-grams whereas for Rouge-2 we calculate the similarity of all 2-grams. Rouge-L refers to the similarity of the longest common subsequence between prediction and target. Important to note for Rouge-L, the grams do not need to be consecutive, but have to be in order. The scores lie

between 0 and 100, where 100 is the best score.

**BLEU [49].** Similar to the Rouge metric, the BLEU score, which is the abbreviation for Bilingual Evaluation Understudy, is used to evaluate the similarity of generated and reference text. To calculate the score, the precision and brevity between the two sentences have to be determined. The precision is the ratio of n-grams that match exactly between generated and reference text. Usually, n goes up to 4. Brevity, on the other hand, penalizes the score of the generated text, if it's shorter than the reference. Combining brevity and precision results in the BLEU score of the generated text. The score itself is again between 0 and 100, where higher scores are better. We use the SacreBLEU [50] version of BLEU.

**Levenshtein Distance.** Lastly, to evaluate PFL-DocVQA, we also use the Levenshtein Distance. This metric is used to directly compare strings on a letter by letter basis. The Levenshtein Distance calculates the minimum amount of substitutions, insertions, and deletions between two sequences. We use the normalized version to have a score between 0 and 100 independent of sequence length. As with the other metrics, the higher the score the better.

## G   Abbreviations

In Table 26, we show the abbreviations we used throughout this paper for the different private in context learning methods of LLMs.

Table 26: **Abbreviations for ICL papers** and their proposed techniques.

| Publication Abbreviation | Publication Name | Technique Abbreviation | Privacy Technique |
| --- | --- | --- | --- |
| DP-ICL | Privacy-Preserving In-Context Learning for Large Language Models [60] | DP-ICL Classification | DP-ICL for text classification |
| | | ESA | Embedding Space Aggregation |
| | | KSA | Keyword Space Aggregation |
| DP-OPT | DP-OPT: Make Large Language Model Your Privacy-Preserving Prompt Engineer [25] | DP-OPT | Differentially-Private Offsite Prompt Tuning |
| FewShotGen | Privacy-Preserving In-Context Learning with Differentially Private Few-Shot Generation [53] | FewShotGen | Differential Private Few-Shot Generation |
| PrivatePrompts [15] | Flocks of Stochastic Parrots: Differentially Private Prompt Learning for Large Language Models [15] | PromptDPSGD [15] | DPSGD for Private Soft Prompt Learning |
| | | PromptPATE [15] | PATE for Privacy-Preserving Discrete Prompts |

