# OpenReview forum: "Open LLMs are Necessary for Current Private Adaptations and Outperform their Closed Alternatives"
_NeurIPS.cc/2024/Conference — NeurIPS 2024 poster_

### Official Review · Reviewer_my5W · 2024-07-04

**Soundness:** 3
**Presentation:** 4
**Contribution:** 3
**Rating:** 7
**Confidence:** 4

**Summary:**

The paper critiques the practices of privately adapting closed(-source) LLMs to private data by demonstrating that these techniques are potentially unsafe and do not yield the required quality of resulting model in terms of accuracy. The authors conclude that a focus on open LLMs should be preferable in sensitive fields.

**Strengths:**

I agree with both the motivations and the premise of this paper. It is important to think about adapting LLMs to private data holistically, i.e. on a systems level, and not just on the level of individual techniques. I like that the authors investigate a multitude of techniques, contrast open and closed LLMs, and discuss the costs of each method. The paper is also written clearly and includes a good overview of prior works. I also approve of the detailed exposition on hyperparameters, costs, etc. in the appendix.

**Weaknesses:**

I would have liked to see a more detailed investigation on the effect of privacy levels beyond $\varepsilon=8$ (for most techniques excluding PATE, where there is a note on different privacy levels and the performance plateauing). Perhaps I missed this, but I didn't find a concrete justification for $\varepsilon=8$ either, although I recognise this as a "folklore/default" privacy budget which many of us have come to think of "automatically" when working with DP-SGD due to the multitude of works which use it.
Moreover, it would have been nice to see more dedicated instruction fine-tuning tasks, since this arguably has become one of the most important tasks for contemporary LLMs.

 On a more conceptual level: There is no "methodological or theoretical novelty" in this work in the traditional sense: It is a (thorough and thoughtful) comparison of techniques. I don't personally view this as a big weakness, to the contrary, I disapprove of the, usually fairly arbitrary, notions of "novelty" used to argue against papers, and I would like to see this paper published in some form, as I believe it discusses important points. However, I am unsure whether it would not have been a better fit to the benchmark track, rather than the main conference track.

**Questions:**

- What are the "privacy units" in this work? In other words, how are adjacent databases defined, and how is this selection justified?
- Privacy agains the LLM provider is a strong argument. I'd be interested in some more discussion on the threat model also in terms of releasing a privately fine-tuned model and the types of attacks which are possible, e.g. MIA or reconstruction attacks, and how they would be carried out. To clarify: I am not asking you to perform any attacks.
- As discussed above, your privacy budget evaluation tops out at 8. I would like to see some discussion on protecting e.g. against reconstruction attacks, which is feasible even with much higher privacy budgets. This should be part of the discussion on potential attacks, and I'd be interested to see when the performance between models converges to each-other and to the non-private performance
- There don't seem to be mixture of experts models involved in your evaluation. Do you believe that this is a limitation of your work? Would you have anticipated similar results and why? I'd welcome a discussion on this point.
- What interactions between privacy and safety did you observe? Do you anticipate that solving a privacy problem could introduce an alignment or safety problem in these models?

**Limitations:**

The authors discuss some limitations, such as cost and performance, which I appreciate. I would be interested (see Questions above) in a discussion on the interplay between privacy and other aspects of trustworthiness, especially safety and alignment.

---

> ### Author Rebuttal · Authors · 2024-08-06
>
> We thank the reviewer for the thorough and thoughtful feedback, we appreciate that. We answer the concerns one by one below:
>
> >**I would have liked to see a more detailed investigation on the effect of privacy levels beyond $\varepsilon=8$ (for most techniques excluding PATE, where there is a note on different privacy levels and the performance plateauing). Perhaps I missed this, but I didn't find a concrete justification for $\varepsilon=8$ either, although I recognise this as a "folklore/default" privacy budget which many of us have come to think of "automatically" when working with DP-SGD due to the multitude of works which use it.**
>
> We fully agree with the reviewer that we need *the investigation on the effect of privacy levels beyond $\varepsilon=8$* and this is what we provided with our privacy-utility trade-off graphs in Figure 2 with $\varepsilon$ ranging from 0 to 8 in the main paper for the classification tasks and the corresponding Figure 3 for the generation tasks (in the Appendix). We observe that across all the privacy budgets ($\varepsilon \in [0,8]$), the privacy-preserving adaptations for open LLMs perform significantly better than the ones for closed LLMs.
>
> >**Private instruction fine-tuning:**
>
> The answer is provided in the general response.
>
> >**On a more conceptual level: There is no "methodological or theoretical novelty" in this work in the traditional sense: It is a (thorough and thoughtful) comparison of techniques. I don't personally view this as a big weakness, to the contrary, I disapprove of the, usually fairly arbitrary, notions of "novelty" used to argue against papers, and I would like to see this paper published in some form, as I believe it discusses important points. However, I am unsure whether it would not have been a better fit to the benchmark track, rather than the main conference track.**
>
> This is a valid concern, and we agree with the reviewer’s perspective on novelty. We would like to note that our paper makes three main contributions, which we outlined at the end of the introduction section:
>
> 1. **Conceptual Analysis**: We provide a comprehensive analysis of privacy-preserving adaptation methods for LLMs.
>
> 2. **Benchmark Evaluation**: We thoroughly evaluate current approaches to private adaptations of LLMs, presenting a detailed benchmark study.
>
> 3. **Proposed Methods**: We introduce PromptDPSGDGen and PromptPATEGen, new differentially private prompts designed for generation tasks. These methods achieve performance levels comparable to LoRA or fine-tuning, as noted in line 68 of our submission (due to formatting, this point might not have been clearly visible under Table 1). Detailed explanations of these techniques can be found in Section 3: “Prompt-based Private Adaptations for Text Generation.”
>
> While the benchmark is one of our primary contributions, we also want to highlight the significance of our conceptual analysis and new methods. We believe these contributions are crucial and justify our submission to the main conference. We hope our work can reach a broader community and further advance privacy-preserving methods for LLMs.
>
> >**Q1: Privacy units.**
>
> We follow the previous work in this domain (e.g., [1,2,3]) and consider the instance-level DP. Two datasets are adjacent if and only if one can be obtained from the other by including an extra record [4] (this definition is used since we track the privacy loss through Rényi Differential Privacy or RDP [5]). For the classification tasks, each record consists of input sentences and a label to be predicted. For the generation tasks, such as the dialog summarization, the record consists of a dialog and its corresponding summary to be predicted.
>
> >**Q2: Threats against privately tuned models.**
>
> *We added the following to Section 2:* “Previous work [3] demonstrated that adapting an LLM with private data is vulnerable to MIAs. However, stronger privacy guarantees result in a lower MIA success rate [3]. The same trend is observed for reconstruction attacks [3].
>
> >**Q3: Convergence between private and non-private training.**
>
> We included extended experiments in Figures 1 and 2 in the attached PDF. Our observations indicate that adapting models with small $\varepsilon \in [0.3,1]$ results in significantly higher performance compared to 0-shot. Additionally, adaptations with $\varepsilon=8$ achieve performance levels close to non-private adaptations.
>
> >**Q4: Evaluation on Mixture of Experts models**
>
> We added the results using PrivateLoRA for Mistral-8x7B to the main paper. We observe similar trends for these models as for the other open LLMs (such as Llama3-8B).
>
> | Method | SST2| Trec | Mpqa | Disaster | Average |
> |-|-|-|-|-|-|
> | PrivateLoRA | 94.5  | 95.8 | 86.6 | 79.2 | 89.02 |
>
> SAMSum:
>
> | Method | Rouge-1 | Rouge-2 | Rouge-L | T($) | Q($) | All($) |
> |-|-|-|-|-|-|-|
> | PrivateLoRA | 52.8$\pm$0.4 | 29.6 $\pm$ 0.2 | 44.7 $\pm$ 0.2 | 57.96 | 9.99 | 67.95 |
>
> MIT-D and MIT-G
>
> | Method | MIT-D | MIT-G | T($) | Q($) | All($) |
> |-|-|-|-|-|-|
> | PrivateLoRA | 93.0  | 69.7  | 1.52 | 9.47 | 10.99  |
>
>
> >**Q5: Interplay of safety and privacy.**
>
> The answer is provided in the general response.
>
> **References:**
>
> [1] “Large Language Models Can Be Strong Differentially Private Learners”. Xuechen Li, Florian Tramer, Percy Liang, Tatsunori Hashimoto. ICLR 2022.
>
> [2] “Differentially Private Fine-tuning of Language Models”. Da Yu, Saurabh Naik, Arturs Backurs, Sivakanth Gopi, Huseyin A. Inan, Gautam Kamath, Janardhan Kulkarni, Yin Tat Lee, Andre Manoel, Lukas Wutschitz, Sergey Yekhanin, Huishuai Zhang. ICLR 2022.
>
> [3] “Flocks of Stochastic Parrots: Differentially Private Prompt Learning for Large Language Models” Haonan Duan, Adam Dziedzic, Nicolas Papernot, Franziska Boenisch. NeurIPS 2023.
>
> [4] “Rényi Differential Privacy of the Sampled Gaussian Mechanism.” Ilya Mironov, Kunal Talwar, Li Zhang. 2019.
>
> [5] “Rényi Differential Privacy”. Ilya Mironov. 2017 IEEE 30th Computer Security Foundations Symposium (CSF).

---

> > ### Comment · Reviewer_my5W · 2024-08-08
> > **Thank you and response to rebuttal.**
> >
> > Thank you for the responses, which I have read carefully. I appreciate the effort you put into the general response, and I would welcome it if these new results found their way into the manuscript, especially the results on safety, which I find intriguing and worthy of inclusion in the main body alongside the results on MoE, with the convergence results perhaps being more appropriate for the appendix (although I'm not trying to micromanage you here, this is just an opinion). Some final comments on my side:
> >
> > - On the topic of privacy budgets beyond $\varepsilon=8$: I meant $\varepsilon > 8$. In particular, since reconstruction attacks can be defended against with very large privacy budgets ($\mathcal{O}(1000)$ and above), it would have been interesting to see some results in this direction too. However, as I recognise that I should have written "larger than" and better specified my request, I will not hold this against you, as I also recognise that this is a topic which is tangental to your work.
> > - On the topic of novelty: I agree with you. The contributions of this paper are strong, and my suggestion for the benchmark track were not meant as a criticism. You have my support for the main conference, especially after the rebuttal.
> > - On the topic of privacy units: Thank you for clarifying this. Please specify it precisely in the final manuscript. A side note in case this may have slipped your attention: The argument you seem to be making about using add one since you are using RDP for accounting does not have a causal basis: RDP can be used with any adjacency notion (e.g. using the Google DP accounting library).
> >
> > In conclusion, I'd like to congratulate you on this nice work. I have increased my score to indicate my clear support for acceptance as is.

---

> ### Author Response · Authors · 2024-08-08
> **Thank you for the prompt response & such positive feedback!**
>
> We thank the Reviewer for the prompt response and are pleased to receive such positive feedback. We also appreciate that the Reviewer supports our work and increased the score.
>
> >**On the topic of privacy budgets beyond $\varepsilon=8$: I meant $\varepsilon>8$. In particular, since reconstruction attacks can be defended against with very large privacy budgets ($O(1000)$ and above), it would have been interesting to see some results in this direction too. However, as I recognise that I should have written "larger than" and better specified my request, I will not hold this against you, as I also recognise that this is a topic which is tangental to your work.**
>
> Thank you for the clarification. The criticism of the previous work was primarily targeting too high $\varepsilon$ values, which led us to concentrate on the $\varepsilon<8$ instead of considering the full rage of the possible $\varepsilon$ values.
>
> We are willing to provide the results on reconstruction attacks for larger privacy budgets. Could the Reviewer please point us out to any previous work that we should/could leverage here? Specifically regarding that “reconstruction attacks can be defended against with very large privacy budgets ($O(1000)$ and above)”.  Would the Reviewer recommend using a specific large value, such as $\varepsilon = 1000$, to evaluate how adapted LLMs perform in defending against such attacks?
>
> >**On the topic of privacy units: Thank you for clarifying this. Please specify it precisely in the final manuscript. A side note in case this may have slipped your attention: The argument you seem to be making about using add one since you are using RDP for accounting does not have a causal basis: RDP can be used with any adjacency notion (e.g. using the Google DP accounting library).**
>
> We appreciate this insight and fully agree that RDP can be used with any adjacency notion. Indeed, an alternative definition of adjacency assumes the two neighboring datasets are of equal size and is based on the replacement of a single record in one of the datasets. We should have been more precise in our explanation during the rebuttal. Our evaluation relied on private-transformers which ​​adopted the definition of *neighboring* based on addition/removal [1], which in turn was taken from an earlier work on RDP [2]. We already specified it precisely in our manuscript.
>
> **References:**
>
> [1] “Large Language Models Can Be Strong Differentially Private Learners”. Xuechen Li, Florian Tramer, Percy Liang, Tatsunori Hashimoto. ICLR 2022 (reference 29 in the main paper).
>
> [2] “Rényi Differential Privacy of the Sampled Gaussian Mechanism.” Ilya Mironov, Kunal Talwar, Li Zhang. 2019 (a new reference).

---

> > ### Comment · Reviewer_my5W · 2024-08-09
> > **On the topic of large $\varepsilon$ values**
> >
> > > Thank you for the clarification. The criticism of the previous work was primarily targeting too high $\varepsilon$ values, which led us to concentrate on the $\varepsilon<8$ instead of considering the full rage of the possible $\varepsilon$ values.
> >
> > Thank you for clarifying, I understand, and agree that the high privacy regime is very important, I thus see the rationale behind reporting the lower values in your current work.
> >
> > > We are willing to provide the results on reconstruction attacks for larger privacy budgets. Could the Reviewer please point us out to any previous work that we should/could leverage here? Specifically regarding that “reconstruction attacks can be defended against with very large privacy budgets ($O(1000)$ and above)”. Would the Reviewer recommend using a specific large value, such as $\varepsilon = 1000$, to evaluate how adapted LLMs perform in defending against such attacks?
> >
> > Let me preface my response by clarifying that my current score is unconditional of any further experiments you may want to conduct, as I think that doing reconstruction attacks in a rebuttal phase with only a few days left is a big ask. Therefore, please regard my recommendation primarily as a stimulus for future work or an add-on to your paper.
> > In case you would like to try to attempt reconstructions (which would probably be elements of the embedding space, not actual inputs, which adds a layer of complexity here), you could look at the work of Hayes et al., NEURIPS 2023 (https://proceedings.neurips.cc/paper_files/paper/2023/file/f8928b073ccbec15d35f2a9d39430bfd-Paper-Conference.pdf). I would like to stress that the findings from such experiments may not be particularly surprising, in the sense that they will probably yield the (already known fact) that "DP with very large epsilon (I've seen values as astronomical as 10^9 in recent works) protects against input (in your case, probably embedding) reconstruction while not significantly impacting utility".

---

> > > ### Author Response · Authors · 2024-08-12
> > > **Reconstruction Attacks**
> > >
> > > We appreciate the Reviewer’s suggestion, which provides valuable direction for future work.
> > >
> > > Given that a few private adaptations for LLMs also utilize the DP-SGD algorithm [2] (e.g., PromptDPSGD[5]), further exploration of the recommended paper [1] will undoubtedly be highly interesting.
> > >
> > > There are also PATE-based [3,4] adaptations, such as PromptPATE [5]. Since these methods rely on public data to train the student prompt, with private data never being used to train any released prompts, reconstruction attacks are likely not feasible.
> > >
> > > **References:**
> > >
> > > [1] “Bounding Training Data Reconstruction in DP-SGD.” Jamie Hayes, Saeed Mahloujifar, Borja Balle. NeurIPS 2023.
> > >
> > > [2] “Deep Learning with Differential Privacy.” Martín Abadi, Andy Chu, Ian Goodfellow, H. Brendan McMahan, Ilya Mironov, Kunal Talwar, Li Zhang. CCS 2016.
> > >
> > > [3] “Semi-supervised Knowledge Transfer for Deep Learning from Private Training Data.” Nicolas Papernot, Martín Abadi, Úlfar Erlingsson, Ian Goodfellow, Kunal Talwar. ICLR 2017.
> > >
> > > [4] “Scalable Private Learning with PATE.” Nicolas Papernot, Shuang Song, Ilya Mironov, Ananth Raghunathan, Kunal Talwar, Úlfar Erlingsson. ICLR 2018.
> > >
> > > [5] “Flocks of Stochastic Parrots: Differentially Private Prompt Learning for Large Language Models.” Haonan Duan, Adam Dziedzic, Nicolas Papernot, Franziska Boenisch. NeurIPS 2023.

---

### Official Review · Reviewer_k2x5 · 2024-07-07

**Soundness:** 2
**Presentation:** 2
**Contribution:** 1
**Rating:** 4
**Confidence:** 4

**Summary:**

This paper compares the performance and privacy leakage between private adaptations on closed and open-source LLMs. The authors conclude that adaptations on open-source LLMs result in better performance, lower training costs, and enhanced privacy protection

**Strengths:**

The paper presents extensive experiments on adaptations of both closed and open-source LLMs, including four private In-Context Learning (ICL) methods for four closed LLMs and three private tuning methods on four open models.

**Weaknesses:**

1. The work lacks novelty in its methods. All the private adaptation methods used are pre-existing, with the authors only extending two methods from classification tasks to generation tasks. This makes the paper more of a benchmark work rather than original research. Additionally, as noted in section 4.2, "Previous work [31] has shown for non-private settings that gradient-based tuning methods (used for open LLMs) offer better accuracy and significantly lower computational costs than ICL (used for closed LLMs) since the adaptations can leverage the internal behavior of the LLM." Thus, it is intuitive that the performance would be similar under DP scenarios, making the conclusion to use open LLMs less impactful.

2. It seems unnecessary to use private tuning methods on open LLMs since the trained model will not be shared and only queried. As the authors state, "Looking at Figure 1, it becomes obvious that any private tuning method executed on that open LLM would, conceptually, improve privacy protection since the LLM provider would neither be involved in the adaptation nor in the use of the adapted LLM, yielding absolute privacy against them." If a company locally fine-tunes the model and then preserves it locally for queries, direct fine-tuning would be more efficient.

3. The comparison in the experiments is unfair and lacks proper baselines. The experiments compare task performance between various adaptations on closed and open LLMs. However, the closed and open models are different and show varying performance even without fine-tuning. The zero-shot performance of these models is not provided, leaving no baseline for reference. Additionally, DP-based ICL methods and fine-tuning methods offer different levels of privacy protection. As the authors mention, "Yet, the threat model of multiple private ICL methods for closed LLMs does not include providing privacy against the LLM provider." It is not fair to compare their performance with the same parameter $\epsilon$ as shown in Figure 2, which could mislead readers into thinking they offer the same level of privacy protection.

**Questions:**

The authors categorize privacy leakage into Private Training Data and Private Query Data. However, I did not see any data inference in the experiments. Is there any data reconstruction attack or metric used to measure the leakage in your model?

**Limitations:**

Please refer to the weaknesses and questions sections.

---

> ### Author Rebuttal · Authors · 2024-08-05
>
> We thank the reviewer for the insightful comments and we are happy that the reviewer appreciates our extensive experiments. We address individual points below one by one:
>
> >**W1: The work lacks novelty.**
>
> We provide a comprehensive privacy analysis and introduce differentially private prompts for generation tasks that, for the first time, achieve performance comparable to LoRA or fine-tuning. The argument for using open LLMs for private adaptations extends beyond their ability to employ gradient-based methods. Our work provides deeper insights into the problem and a thorough overview of the field. Additionally, methods proposed for closed LLMs can be applied to open LLMs. Notably, our PromptPATEGen (an ICL-based approach) applied to open LLMs outperforms other private adaptations of closed LLMs, as evidenced, e.g., in Table 4.
>
> >**W2: It seems unnecessary to use private tuning methods on open LLMs since the trained model will not be shared and only queried.**
>
> Please refer to Figure 1 in the submission, where we explain that the answers to the queries leak information about the private data used to create the prompts. Note that the previous work [1, 2, and 3] demonstrated that the data used to create the prompts can leak to the querying party, e.g., by leveraging the membership inference attacks (the reviewer also asked for such attacks in the Question below).
>
> >**W3: The comparison in the experiments is unfair and lacks proper baselines. The experiments compare task performance between various adaptations on closed and open LLMs. However, the closed and open models are different and show varying performance even without fine-tuning. The zero-shot performance of these models is not provided, leaving no baseline for reference.**
>
> The zero-shot performance was provided in the cited papers. We also included the required numbers in the Table below and updated Tables 3, 4, 5, and 6 in the main paper accordingly. Please also refer to Figures 2 and 3 in the attached PDF, where we present the performance for zero-shot and the $\varepsilon=\infty$ as baselines. Our observations indicate that adapting models with small $\varepsilon \in [0.3,1]$ results in significantly higher performance compared to zero-shot. Additionally, adaptations with $\varepsilon=8$ achieve performance levels close to non-private adaptations. Overall, while the zero-shot performance is higher for the closed LLMs, the private adaptations perform better on open LLMs than on closed ones.
>
> Table: Comparison between closed vs open LLMs with baselines: zero-shot and $\varepsilon=\infty$.
>
> |Method|Model|SST2|Trec|Mpqa|Disaster|Average|
> |---|---|---|---|---|---|---|
> |zero-shot|GPT3-Davinci closed|92.4$\pm$0.0|51.8$\pm$0.2|84.5$\pm$0.1|76.4$\pm$0.2|76.3|
> |$\varepsilon=8$ DP-OPT|GPT3-Davinci closed|92.2$\pm$0.8|68.7$\pm$6.5|85.8$\pm$0.7|78.9$\pm$0.3|81.4|
> |$\varepsilon=\infty$ ICL|GPT3-Davinci closed|94.7$\pm$0.4|79.1$\pm$0.5|88.8$\pm$0.1|69.0$\pm$5.9|82.9|
> |zero-shot|Vicuna 7B open|85.8|47|78.8|56.7|67.1|
> |$\varepsilon=8$ PrivateLoRA|Vicuna 7B open|96.0$\pm$0.1|96.8$\pm$0.2|87.3$\pm$0.2|80.8$\pm$0.1|90.2|
> |$\varepsilon=\infty$ LoRA|Vicuna 7B open|96.4|98.2|87.5|82.1|91.1|
>
> >**It is not fair to compare their performance with the same parameter $\varepsilon$ as shown in Figure 2 which could mislead readers into thinking they offer the same level of privacy protection.**
>
> We show in Figure 1 that *privacy leakage types* A (the data owner’s private data leaks to the LLM provider) and B (the private query of the querying party leaks to the LLM provider) are inherent to closed LLM and are not protected by any of the tested methods. Thus, we measured the privacy leakage using the DP framework only for case C (where private information from the data owner leaks to the querying party). Even under this most favorable scenario for closed LLMs, they underperform compared to open LLMs when used for private adaptations.
>
> >**The authors categorize privacy leakage into Private Training Data and Private Query Data. However, I did not see any data inference in the experiments. Is there any data reconstruction attack or metric used to measure the leakage in your model?**
>
> **Private Training Data:** Previous work [2, 3 (Table 8)] demonstrated that applying (stronger) privacy guarantees reduces privacy leakage. In our submission, Figures 2 and 3 illustrate the performance depending on the privacy budgets $\varepsilon$ ranging from 0 to 8. The lower the $\varepsilon$, the lower the leakage (better protection against membership inference or reconstruction attacks).
>
> **Private Query Data:** For closed LLMs, queries are sent directly to the LLM provider in plain form, resulting in clear privacy leakage. In contrast, with open LLMs, the query is only shared with a data owner, such as a hospital or a bank, which already has access to private data. Thus, the private adaptations via open LLMs minimize the exposure of sensitive information.
>
> **References:**
>
> [1] “On the privacy risk of in-context learning”. Haonan Duan, Adam Dziedzic, Mohammad Yaghini, Nicolas Papernot, Franziska Boenisch. In The 61st Annual Meeting Of The Association For Computational Linguistics 2023.
>
> [2] “Flocks of Stochastic Parrots: Differentially Private Prompt Learning for Large Language Models”. Haonan Duan, Adam Dziedzic, Nicolas Papernot, Franziska Boenisch. In the Thirty-seventh Conference on Neural Information Processing Systems (NeurIPS) 2023.
>
> [3] "DP-OPT: Make Large Language Model Your Privacy-Preserving Prompt Engineer". Junyuan Hong, Jiachen T. Wang, Chenhui Zhang, Zhangheng Li, Bo Li, Zhangyang Wang. ICLR 2024.
>
> ---
> Once more, we thank the reviewer for the constructive feedback. Addressing the comments has improved the quality of our work. We look forward to further discussions and continued advancements in this evolving field. Please let us know if we can provide any additional information to further enhance your assessment of our work and potentially increase the score.

---

> > ### Comment · Reviewer_k2x5 · 2024-08-08
> > **Response to authors' rebuttal**
> >
> > Thank you for your detailed experiments and explanations. While some of my concerns have been addressed, I still have two main points regarding W3 and Q1 that require further clarification:
> >
> > 1.
> > I appreciate the use of Differential Privacy (DP) as a framework for measuring privacy and providing guarantees. However, I'm concerned about the comparison setting in your experiments. Given that the closed LLM uses Prompt Tuning (PT) while the open LLM uses model Fine-Tuning (FT), there's insufficient evidence to suggest that applying the same DP parameter $\epsilon$ to both PT and FT provides equivalent privacy guarantees for the training data. Therefore, using the same ε across different scenarios may not be appropriate for a fair comparison.
> >
> > 2.
> > The use of DP ε as the sole indicator of privacy leakage level is not fully convincing. As mentioned in the first point, PT and FT may have different privacy risks under the same $\epsilon$. To provide a more comprehensive evaluation of privacy leakage for both Private Training Data and Private Query Data, I suggest incorporating additional attack methods, such as membership inference attacks or model inversion attacks.

---

> ### Author Response · Authors · 2024-08-08
> **Thank you for the prompt response & positive feedback!**
>
> We appreciate the Reviewer’s prompt response and the willingness to help us further improve our submission. We are also happy that we were already able to address most of the Reviewer’s concerns.
>
> >**1. I appreciate the use of Differential Privacy (DP) as a framework for measuring privacy and providing guarantees. However, I'm concerned about the comparison setting in your experiments. Given that the closed LLM uses Prompt Tuning (PT) while the open LLM uses model Fine-Tuning (FT), there's insufficient evidence to suggest that applying the same DP parameter $\varepsilon$ to both PT and FT provides equivalent privacy guarantees for the training data. Therefore, using the same $\varepsilon$ across different scenarios may not be appropriate for a fair comparison.**
>
> Comparing different methods at the same privacy budget $\varepsilon$ is a standard approach and was used in the previous work. For example, in [1], Table 2 presents the comparison between DP-OPT (which uses hard prompts) and PromptDPSGD (which employs soft-prompt tuning) at the same privacy level $\varepsilon=8$ and the same parameter $\delta=1/|D|$, where $|D|$ denotes the size of the fine-tuning set. Therefore, we believed that this comparison setting was justified.
>
> >**2. The use of DP $\varepsilon$ as the sole indicator of privacy leakage level is not fully convincing. As mentioned in the first point, PT and FT may have different privacy risks under the same $\varepsilon$. To provide a more comprehensive evaluation of privacy leakage for both Private Training Data and Private Query Data, I suggest incorporating additional attack methods, such as membership inference attacks or model inversion attacks.**
>
> We understand the Reviewer’s concerns. This is also why, in the rebuttal, we referred to previous work [1 (Table 8), 2,3 (Figure 2)] that analyzed the privacy leakage of the LLM adaptations using Membership Inference Attacks (MIAs).
>
> To clarify, is the goal to compare the vulnerability of the ICL (In-Context Learning) methods, used to adapt closed LLMs, with the tuning methods (such as fine-tuning, LoRA, or prefix-tuning which rely on access to model gradients), used for open LLMs, in the context of attacks like membership inference?
>
> We are willing to run an additional experiment that can address these concerns. One of the candidates is to use the ICL method (i.e., hard prompts) and the tuning method (e.g., LoRA) on the same model (e.g., Vicuna 7B) and then compare the success rate of a Membership Inference Attack (MIA) against these two adaptations. These can be executed for the non-private setting as well as for both methods trained under Differential Privacy (with, e.g., $\varepsilon=8$). If the Reviewer meant another evaluation, we would be grateful for further clarification in the comments below.
>
> **References:**
>
> [1] DP-OPT: Make Large Language Model Your Privacy-Preserving Prompt Engineer
> Junyuan Hong, Jiachen T. Wang, Chenhui Zhang, Zhangheng Li, Bo Li, Zhangyang Wang. ICLR 2024 (spotlight).
>
> [2] “On the privacy risk of in-context learning”. Haonan Duan, Adam Dziedzic, Mohammad Yaghini, Nicolas Papernot, Franziska Boenisch. In The 61st Annual Meeting Of The Association For Computational Linguistics 2023.
>
> [3] “Flocks of Stochastic Parrots: Differentially Private Prompt Learning for Large Language Models”. Haonan Duan, Adam Dziedzic, Nicolas Papernot, Franziska Boenisch. In the Thirty-seventh Conference on Neural Information Processing Systems (NeurIPS) 2023.

---

> > ### Author Response · Authors · 2024-08-12
> > **Membership Inference Attacks**
> >
> > We ran the additional experiment to address the concerns. We used the ICL method (i.e., hard prompts with PromptPATE) and LoRA on Vicuna 7B. We executed the experiment for the non-private setting ($\varepsilon=\infty$) as well as for both methods trained under Differential Privacy (with, $\varepsilon=8$ for LoRA and $\varepsilon=0.4$ for PromptPATE, which plateaus after this value and the same outcome is obtained when we set $\varepsilon=8$). We followed the experiment from [1] (Figure 2). Our results indicate that the members and non-members are easily distinguishable for non-private adaptations. In contrast, after running the private adaptations, the distribution probabilities of members vs non-members become significantly less distinguishable which makes the membership inference attacks less successful.
> >
> > If the Reviewer meant another evaluation, we would be grateful for further clarification in the comments below and are willing to answer any further questions.
> >
> > **References:**
> >
> > [1] “Flocks of Stochastic Parrots: Differentially Private Prompt Learning for Large Language Models.” Haonan Duan, Adam Dziedzic, Nicolas Papernot, Franziska Boenisch. NeurIPS 2023.

---

> > > ### Comment · Reviewer_k2x5 · 2024-08-13
> > > **I have raised the score and still concerned about the novelty, DP comparison, and lack of privacy attack**
> > >
> > > Thank you for your diligent work and detailed explanation. I appreciate the effort you've put into this paper. After careful consideration, I have a few thoughts I'd like to share:
> > >
> > > Regarding novelty: While the extension of the two methods to the generation task is interesting, I wonder if we could further explore how this contributes to the field beyond what might be intuitively expected. For instance, the conclusion that fine-tuning outperforms prompt tuning even with DP added seems to align with general expectations.
> > >
> > > On the membership inference attack: This is a valuable addition to the paper. To strengthen this aspect, have you considered incorporating additional types of attacks? This could provide a more comprehensive view of privacy levels beyond the DP noise.
> > >
> > > Concerning the DP comparison: The application of DP to both model fine-tuning and prompt tuning is noteworthy. However, Even though previous work compare the soft and hard prompt tuning under same DP noise, it is different from model finetuning and prompt tuning. It is hard to ensure a fair comparison between these two approaches, given their inherent differences.
> > >
> > > Overall, your paper demonstrates thorough experimentation and is well-written. To further enhance its impact, you might consider addressing the points above, particularly focusing on highlighting the novelty of your approach, expanding the range of privacy attacks examined, and refining the DP comparison methodology.
> > > Thank you again for your hard work.

---

> ### Author Response · Authors · 2024-08-14
> **Novelty, Expectations, and Previous Work**
>
> >**Thank you for your diligent work and detailed explanation. I appreciate the effort you've put into this paper. After careful consideration, I have a few thoughts I'd like to share:**
>
> We greatly appreciate the Reviewer's thoughtful engagement in the discussion period and the careful consideration of our research.
>
> >**Regarding novelty: While the extension of the two methods to the generation task is interesting, I wonder if we could further explore how this contributes to the field beyond what might be intuitively expected.**
>
> Thank you for recognizing the significance of our two novel methods for the generation tasks. *This was not intuitively expected.* Please check Lines 56 to 60 in our submitted manuscript where we explain: “We demonstrate how to locally apply privacy-preserving prompt-based methods to train generation tasks with high-performance -- **claimed impossible by prior work [ 29 ]**. In particular, we show for the first time that private prompt tuning for text generation tasks can achieve comparable performance to private (full) fine-tuning and private low-rank adaptations (LoRA).”
>
> We further write in Lines 138 to 141: “Having prompt-based generation holds the advantage that, in contrast to fine-tuning based approaches, they support mixed-task inference [24, 27, 31], i.e., they require one frozen model for multiple tasks rather than a separate model copy for each of them. This reduces storage and offers greater flexibility and efficiency.”
>
> Our main message emphasizes the importance of developing new adaptation methods for open LLMs, particularly focusing on PEFT methods, especially those that are input-based, like prefix tuning. We would appreciate any further suggestions from the reviewer on how we might strengthen the description of how our methods contribute to the field."
>
> >**For instance, the conclusion that fine-tuning outperforms prompt tuning even with DP added seems to align with general expectations.**
>
> This conclusion is also not aligned with general expectations. The gradient based adaptations (e.g., PromptDPSGD, PrivateLoRA, DP-FineTune) are typically applied to much smaller open LLMs, with models like Llama3-8B containing only up to 8 billion parameters. In contrast, the hard-prompt based methods (e.g., PromptPATE, DP-OPT, DP-FewShotGen) are used with significantly larger LLMs, such as GPT3.5 Davinci with 175B parameters, Claude with 200B parameters (Claude), or even larger models like GPT4-Turbo with 1.76T parameters! The closed models also have much higher zero-shot performance than the open LLMs. However, the adaptation of open LLMs is more powerful, ultimately leading to greater performance gains for specific downstream tasks.
>
> Furthermore, [24] claimed the opposite to our results. Specifically, Table 2 in [24] shows that DP-OPT on GPT3.5 Davinci outperforms PromptDPSG [15] on Vicuna 7B. We observe that this is because of the insufficient fine-tuning of the PromptDPSGD method and without further analysis of other private gradient-based adaptation methods.
>
> Additionally, we would like to address the previous concerns regarding other related work [31] in more detail below:
>
> >**"Previous work [31] has shown for non-private settings that gradient-based tuning methods (used for open LLMs) offer better accuracy and significantly lower computational costs than ICL (used for closed LLMs) since the adaptations can leverage the internal behavior of the LLM." Thus, it is intuitive that the performance would be similar under DP scenarios, making the conclusion to use open LLMs less impactful.**
>
> Note that the conclusion (cited above) from [31] is based only on their single Table 1, which includes only a single gradient-based method ($IA^3$) and solely few-shot ICL, both tested exclusively on held-out tasks from T0. For the gradient-based method, they used T0-11B which has better 0-shot performance than GPT3 Da Vinci on the tasks held out from T0.  One of the Reviewers of the paper [31] wrote: “The paper achieves strong few-shot results with moderate language model sizes (up to 11B), outperforming more expensive models like GPT3.”  However, the fact that $IA^3$ performs better is not surprising.
> In contrast, **our results are striking**, as highlighted by Reviewer gGr3. We analyzed a broader range of models, datasets, and methods, with a specific focus on privacy-preserving adaptations. Firstly, we show that even larger models like GPT-4 underperform compared to open LLMs like Llama3-8B when adapted, despite the open LLMs starting from a much lower zero-shot performance baseline. Secondly, our results not only consider performance and cost, as in [31], but also emphasize the critical aspect of privacy protection for end-users of LLMs.

---

> ### Author Response · Authors · 2024-08-14
> **Membership Inference Attacks & DP Comparison**
>
> >**On the membership inference attack: This is a valuable addition to the paper. To strengthen this aspect, have you considered incorporating additional types of attacks? This could provide a more comprehensive view of privacy levels beyond the DP noise.**
>
> Yes, we have considered incorporating additional attacks, including, for example, the exploration of reconstruction attacks. However, as [noted by Reviewer my5W](https://openreview.net/forum?id=Jf40H5pRW0&noteId=oPC5WsBRPB): “I think that doing reconstruction attacks in a rebuttal phase with only a few days left is a big ask.”
>
> Given that a few private adaptations for LLMs also utilize the DP-SGD algorithm [2] (e.g., PromptDPSGD[5]), further exploration of the recommended paper on reconstruction attacks [1] will undoubtedly be highly interesting.
>
> There are also PATE-based [3,4] adaptations, such as PromptPATE [15]. Since these methods rely on public data to train the student prompt, with private data never being used to train any released prompts, reconstruction attacks are likely not feasible.
>
> >**Concerning the DP comparison: The application of DP to both model fine-tuning and prompt tuning is noteworthy. However, even though previous work compare the soft and hard prompt tuning under same DP noise, it is different from model finetuning and prompt tuning. It is hard to ensure a fair comparison between these two approaches, given their inherent differences.**
>
> We appreciate the Reviewer's concerns regarding fair comparison and understand the emphasis placed on this issue. However, we want to reassure the Reviewer that using differential privacy (DP) as an indicator of privacy leakage is a standard and widely accepted approach, as demonstrated in numerous studies, including [24] which we referenced earlier.
>
> To further support our approach, we direct the Reviewer to Table 2 in [5], where a comparison is made between full fine-tuning, LoRA, prefix-tuning, and other methods, all evaluated at the same privacy budgets of $\varepsilon \in {3,8}$. Additionally, Table 10 in [6] provides a comparison between full fine-tuning and LoRA at a privacy budget of $\varepsilon = 6.7$. These references illustrate that our approach aligns with established practices in the field.
>
> >**Overall, your paper demonstrates thorough experimentation and is well-written.**
>
> We thank the Reviewer for this positive feedback.
>
> >**To further enhance its impact, you might consider addressing the points above, particularly focusing on highlighting the novelty of your approach, expanding the range of privacy attacks examined, and refining the DP comparison methodology.**
>
> We hope that the comments above clarify that we have already addressed the novelty of our approach in our submission. However, if the Reviewer has specific suggestions for improving the presentation of our methods and results, we would be more than happy to consider this feedback. We agree that analyzing privacy attacks beyond membership inference attacks (MIAs) is valuable, and we are actively exploring reconstruction attacks. As previously mentioned, using differential privacy (DP) for comparison is a standard practice in the literature, and we have adhered to this protocol to ensure a fair comparison.
>
> >**Thank you again for your hard work.**
>
> Once again, we thank the reviewer for the constructive feedback on our work. Addressing the comments has significantly improved the quality of our paper. We hope that the answers address the reviewer's concerns and that the scores can be increased.

---

> > ### Author Response · Authors · 2024-08-14
> > **References**
> >
> > [1] “Bounding Training Data Reconstruction in DP-SGD.” Jamie Hayes, Saeed Mahloujifar, Borja Balle. NeurIPS 2023.
> >
> > [2] “Deep Learning with Differential Privacy.” Martín Abadi, Andy Chu, Ian Goodfellow, H. Brendan McMahan, Ilya Mironov, Kunal Talwar, Li Zhang. CCS 2016.
> >
> > [3] “Semi-supervised Knowledge Transfer for Deep Learning from Private Training Data.” Nicolas Papernot, Martín Abadi, Úlfar Erlingsson, Ian Goodfellow, Kunal Talwar. ICLR 2017.
> >
> > [4] “Scalable Private Learning with PATE.” Nicolas Papernot, Shuang Song, Ilya Mironov, Ananth Raghunathan, Kunal Talwar, Úlfar Erlingsson. ICLR 2018.
> >
> > [5] “Large Language Models Can Be Strong Differentially Private Learners.” Xuechen Li, Florian Tramèr, Percy Liang, Tatsunori Hashimoto. ICLR 2022.
> >
> > [6] “Differentially Private Fine-tuning of Language Models.” Da Yu, Saurabh Naik, Arturs Backurs, Sivakanth Gopi, Huseyin A. Inan, Gautam Kamath, Janardhan Kulkarni, Yin Tat Lee, Andre Manoel, Lukas Wutschitz, Sergey Yekhanin, Huishuai Zhang. ICLR 2022.
> >
> > (The references below are aligned with the numbers in the submitted paper.)
> >
> > [15] “Flocks of stochastic parrots: Differentially private prompt learning for large language models.” Haonan Duan, Adam Dziedzic, Nicolas Papernot, and Franziska Boenisch. NeurIPS 2023.
> >
> > [24] “DP-OPT: Make Large Language Model Your Privacy-Preserving Prompt Engineer.” Junyuan Hong, Jiachen T. Wang, Chenhui Zhang, Zhangheng Li, Bo Li, Zhangyang Wang. ICLR 2024.
> >
> > [27] “Prefix-tuning: Optimizing continuous prompts for generation.” Xiang Lisa Li and Percy Liang. ACL 2021.
> >
> > [31] “Few-Shot Parameter-Efficient Fine-Tuning is Better and Cheaper than In-Context Learning”. Haokun Liu, Derek Tam, Muqeeth Mohammed, Jay Mohta, Tenghao Huang, Mohit Bansal, Colin Raffel. NeurIPS 2022. https://openreview.net/pdf?id=rBCvMG-JsPd

---

### Official Review · Reviewer_gGr3 · 2024-07-12

**Soundness:** 2
**Presentation:** 3
**Contribution:** 2
**Rating:** 5
**Confidence:** 4

**Summary:**

The paper compares private adaptation between closed LLMs and open LLMs, and the authors find that adapted open LLMs always perform better than closed ones at much lower cost.

**Strengths:**

1. This paper presents a comprehensive overview of privacy-preserving adaptation techniques for LLMs. It thoroughly examines various existing methods and compares them across key factors like performance and cost.
2. The results are striking: open-source LLMs consistently outperform their counterparts in virtually every aspect.
3. The paper also discusses the private data flowing and point out that the private data might still be leaked to LLM providers in 3 out of 4 adaptation methods studied.

**Weaknesses:**

My main concern is that the claim that open LLMs are essential for achieving high-quality, privacy-preserving adaptation might be premature. It could be that the field is still new, and we simply haven't yet developed sufficiently powerful algorithms for privacy-preserving adaptation of closed models. I suggest tuning down this claim slightly, acknowledging that the field is relatively new and rapidly evolving, and that the current findings may only be applicable to the privacy algorithms currently available.

**Questions:**

Please tune down the claim by adding a remark that the finding only applies to existing methods.

**Limitations:**

As above

---

> ### Author Rebuttal · Authors · 2024-08-06
>
> We appreciate the positive, encouraging, and constructive feedback. We are pleased that the reviewer recognizes our thorough analysis and finds the results *"striking"*. We address the main concern below:
>
> >**My main concern is that the claim that open LLMs are essential for achieving high-quality, privacy-preserving adaptation might be premature. It could be that the field is still new, and we simply haven't yet developed sufficiently powerful algorithms for privacy-preserving adaptation of closed models. I suggest tuning down this claim slightly, acknowledging that the field is relatively new and rapidly evolving and that the current findings may only be applicable to the privacy algorithms currently available. (...) Please tune down the claim by adding a remark that the finding only applies to existing methods.**
>
> Thank you for sharing your concern. We greatly appreciate your feedback. As requested, we have toned down the statements and provided the revised sentences below.
>
> Specifically, we changed the title to: “Open LLMs are Necessary for **Current** Private Adaptations and Outperform their Closed Alternatives”.
>
> We also added the following statement at the end of the abstract: **“It is important to note that the field of privacy-preserving adaptations of LLMs is relatively new and rapidly evolving. Consequently, our findings are based on the currently available methods and may evolve as new techniques are developed.”**
>
> To elaborate on our purpose, we assess the state-of-the-art methods for privacy-preserving adaptations of LLMs based on all the papers in this domain (starting from ICLR submissions in 2022 and up until this submission deadline). Our evaluation also includes the latest models, such as Llama 3 (released on April 14th, 2024, which was shortly before the NeurIPS deadline) and GPT-4. Additionally, we have developed new techniques to enhance the private adaptations of LLMs. Our primary goal is to identify the current strengths and weaknesses of the private adaptations and provide better methods, especially for users who would like to leverage LLMs on their private data. We do develop new private adaptations in this submission to support generation tasks, such as dialogue summarization, and provide insights to help the community advance in this field. Taking into account the fact that there are at least 7 different methods for private adaptations, we contribute 2 more and point to the currently most promising directions.
>
> **The statements that already indicate the consideration of only the current methods are bolded:**
> - We already stated in the abstract the following: (starting from Line 5) In this work, we analyze the privacy protection and performance of **the four most recent methods** for private adaptation of closed LLMs.
>
> **The updated parts of the statements in the submission are bolded:**
>
> - At the end of the abstract (starting from Line 19):  This yields the conclusion that to achieve truly privacy-preserving LLM adaptations that yield high performance and more privacy at lower costs, **taking into account current methods and models**, one should use open LLMs.
> - Starting from Line 51: “Overall, our results indicate that from the perspective of effective privacy-preservation, **current** adaptations of open LLMs are strictly preferable over their closed LLM alternatives, since they are more private, more performant, and less expensive.”
> - Line 45: Our results provide the following insights: (1) All **currently available** methods for adapting closed LLMs leak private query data (intended for the data owner) at inference time to the LLM provider.
> Starting from Line 70: Our extensive experiments on various open and closed LLMs and multiple classification and generation tasks show that the local (gradient-based) adaptations outperform their **current** closed (discrete prompt-based) counterparts in terms of privacy, performance, and cost efficiency.
> - Line 191: DP does not aim at protecting query data. Hence, none of the **current** private ICL methods attempt to protect that data against the LLM provider.
> - Line 333 onward (in Section 5: “Discussion and Future Work”): On the contrary, the leakage of private query data to the LLM provider is **to date** an inherent problem with closed LLMs, since no methods to provide formal guarantees for the query data are **currently** known.
> - Line 351: “**We hope that implementing the above-mentioned solutions will shrink the gap between private adaptations of open and closed LLMs.**”
>
> ---
>
> Once more, we thank the reviewer for the constructive feedback. Addressing the concern has improved the quality of our work. We look forward to further discussions and continued advancements in this evolving field. Please let us know if we can provide any additional information to further enhance your assessment of our work and potentially increase the score.

---

> > ### Comment · Reviewer_gGr3 · 2024-08-07
> > **reply to rebuttal**
> >
> > thanks for the rebuttal. please make sure the changes in the rebuttal is integrated into the camera-ready version. I adjusted my score accordingly.

---

> > > ### Author Response · Authors · 2024-08-08
> > > **Thank you!**
> > >
> > > Thank you for your prompt response and for increasing the score, we appreciate it. We had already incorporated the changes into the paper before submitting the rebuttal.

---

> > > > ### Author Response · Authors · 2024-08-12
> > > > **Have concerns been addressed?**
> > > >
> > > > We are glad that our updated statements addressed the concerns. Since the rating is still on the borderline, we are more than happy to provide any additional insights if needed.

---

### Official Review · Reviewer_1UUG · 2024-07-17

**Soundness:** 3
**Presentation:** 3
**Contribution:** 3
**Rating:** 7
**Confidence:** 2

**Summary:**

This paper compares and contrasts the privacy protections of open vs closed LLMs through conceptual threat models and experimental evaluation. The authors give experimental evidence on local gradient based adaptations performing better than their closed discrete prompt-based counterparts in the areas of privacy, cost-efficiency and performance. The paper concludes that the use of open LLMs not only yields more privacy, but also higher performance at lower costs.

**Strengths:**

The paper is well-written and clearly explains what the contributions are, the background for both DP and LLMs, as well as their experimental evidence. The authors make a compelling case behind the reasons for privacy-preserving adaptations for closed LLMs not being as effective. The problem addressed in the paper is important and the contribution seems novel.

**Weaknesses:**

I think the paper is well-written and don't have any specific feedback.

**Questions:**

N/A

**Limitations:**

Limitations were addressed

---

> ### Author Rebuttal · Authors · 2024-07-31
>
> >**The paper is well-written and clearly explains what the contributions are, the background for both DP and LLMs, as well as their experimental evidence. The authors make a compelling case behind the reasons for privacy-preserving adaptations for closed LLMs not being as effective. The problem addressed in the paper is important and the contribution seems novel. I think the paper is well-written and don't have any specific feedback. Limitations were addressed.**
>
> Thank you for the positive feedback and insightful comments. We greatly appreciate your review. Please let us know if there are any additional steps or information we can provide to further enhance your confidence in the assessment of our work.

---

### Author Rebuttal · Authors · 2024-08-06

We would like to thank all the reviewers for their valuable feedback and insightful comments, which greatly helped us further improve our submission. Our results were described as “striking” by Reviewer gG3r. This work contains “extensive experiments” (Reviewer k2x5), which demonstrate that adaptations of open LLMs outperform their closed counterparts “in the areas of privacy, cost-efficiency, and performance” (Reviewer 1UUG). Additionally, it was recognized that “the problem addressed in the paper is important” (Reviewer 1UUG) and that “this paper should be published” (Reviewer my5w). It is important to think about adapting LLMs to private data holistically, i.e. on a systems level, and not just on the level of individual techniques (Reviewer my5w). We hope that our work can significantly contribute to the community and reach a broader audience.

Below, we present new experimental findings that we believe are valuable to share with all reviewers. Following this, we provide individual responses addressing each reviewer's comments in detail.

>**Interplay of safety and privacy: What interactions between privacy and safety did you observe? Do you anticipate that solving a privacy problem could introduce an alignment or safety problem in these models?**

We conducted additional experiments to analyze how fine-tuning a downstream task with and without differential privacy affects the safety alignment of models. We followed the approach from [1] to evaluate Mixtral-8x7B-instruct and fine-tuned the model on SAMSum once with $\varepsilon=8$ and once with $\varepsilon=\infty$. We selected SAMSum as it does contain conversations with unsafe language (e.g., cursing or harassment). These results are presented in Table 1 in the attached PDF (and also shown below for convenience). The table is divided into two sections, “Compliance on Harmful Queries” and “Refusal on Harmless Queries”. The scores were generated by separately prompting our model with 100 harmful and 100 harmless queries, each repeated 20 times with different safety prompts. Finally, the outputs are categorized as complying or refusing the input by Llama Guard, giving the percentage of incorrectly handled answers in our table. Therefore, the lower the score, the better.

First, we observe that fine-tuning of any kind decreases the model's safety capability. This is easily identifiable in the table, as the original model has lower scores across the board for compliance with harmful behavior compared to both fine-tuned models. Similar results were also shown in [2] and [3], where the authors used a dataset containing unsafe samples to fine-tune safety-aligned models, which drastically increases compliance with unsafe behavior.

Second, **we can observe that private fine-tuning affects the safeguards less than non-private fine-tuning**. In the non-private case, the influence of individual samples is unrestricted, which can amplify the impact of unsafe samples, as shown in [2] and [3]. In contrast, differential privacy (DP) limits the influence of any single sample. Consequently, the impact of unsafe examples is minimized, resulting in a model that retains more of its safe behavior.

||Harmful|||||Harmless||||
|-|-|-|-|-|-|-|-|-|-|
||noprompt|default|mistral|short||noprompt|default|mistral|short|
|Basemodel|36|0|0|1||0|2|0|1|
|LoRA|64|39|30|38||0|3|2|2|
|PrivateLoRa|48|7|5|9||0|1|0|2|

>**Private instruction fine-tuning: it would have been nice to see more dedicated instruction fine-tuning tasks since this arguably has become one of the most important tasks for contemporary LLMs.**

Indeed, we also agree that instruction fine-tuning (IFT) tasks should be executed with privacy.

In general, users send instructions to an LLM and these instructions are collected by the LLM provider. The LLM provider later uses annotators to select and annotate (e.g., provide answers to) the instructions, which are subsequently used to further improve the instruction following capabilities of the LLM. The private instructions might leak to (1) the annotators and then to (2) the other users interacting with the LLM.

To evaluate the performance of IFT with privacy, we use the Pythia 1B model fine-tuned on the AlpaGasus-9k dataset [4] (annotated and filtered with an LLM). We use Private LoRA for the fine-tuning. The results are shown in Figure 3 of our attached PDF. We compare the model outputs against various instruction sets, where these outputs were scored by GPT4-Turbo. A model wins if its score is higher than the other model’s score, whereas they draw if they have the same score. We observe that while the model trained without privacy ($\varepsilon=\infty$) has more wins than the model trained with privacy ($\varepsilon=8$), the majority of outputs were scored equally between the two models. This indicates that **the instruction fine-tuning tasks can also be effectively addressed with privacy protection**.

**References:**

[1] “On Prompt-Driven Safeguarding for Large Language Models". Chujie Zheng, Fan Yin, Hao Zhou, Fandong Meng, Jie Zhou, Kai-Wei Chang, Minlie Huang, Nanyun Peng. ICML 2024.

[2] "LoRA Fine-tuning Efficiently Undoes Safety Training in Llama 2-Chat 70B". Simon Lermen, Charlie Rogers-Smith, Jeffrey Ladish. ICLR 2024 Workshop.

[3] "Learning and Forgetting Unsafe Examples in Large Language Models". Jiachen Zhao, Zhun Deng, David Madras, James Zou, Mengye Ren. ICML 2024.

[4] “AlpaGasus: Training a Better Alpaca with Fewer Data”. Lichang Chen, Shiyang Li, Jun Yan, Hai Wang, Kalpa Gunaratna, Vikas Yadav, Zheng Tang, Vijay Srinivasan, Tianyi Zhou, Heng Huang, Hongxia Jin. ICLR 2024.

---

Once again, we thank the reviewers for the constructive feedback. Addressing the concerns has improved the quality of our work. We look forward to further discussions and continued advancements in this evolving field. Please let us know if we can provide any additional information to further enhance your assessment of our work.

---

### Decision · Program_Chairs · 2024-09-25

**Decision:**

Accept (poster)

**Comment:**

The paper offers a thoughtful critique of using closed-source LLMs to perform private learning, noting that most of these methods actually leak information to the LLM provider, and instead argues that fine-tuning an open-source LLM is preferable from both a privacy and utility perspective. All reviewers had positive things to say about the paper and appreciated its contributions.